JCB Journal of Cell Biology

# PIGB maintains nuclear lamina organization in skeletal muscle of *Drosophila*

Miki Yamamoto-Hino[1], Masaru Ariura[2], Masahito Tanaka[3], Yuka W. Iwasaki[2,4,5], Kohei Kawaguchi[1], Yuta Shimamoto[3], and Satoshi Goto[1]

The nuclear lamina (NL) plays various roles and participates in nuclear integrity, chromatin organization, and transcriptional regulation. Lamin proteins, the main components of the NL, form a homogeneous meshwork structure under the nuclear envelope. Lamins are essential, but it is unknown whether their homogeneous distribution is important for nuclear function. Here, we found that PIGB, an enzyme involved in glycosylphosphatidylinositol (GPI) synthesis, is responsible for the homogeneous lamin meshwork in *Drosophila*. Loss of *PIGB* resulted in heterogeneous distributions of B-type lamin and lamin-binding proteins in larval muscles. These phenotypes were rescued by expression of PIGB lacking GPI synthesis activity. The *PIGB* mutant exhibited changes in lamina-associated domains that are large heterochromatic genomic regions in the NL, reduction of nuclear stiffness, and deformation of muscle fibers. These results suggest that PIGB maintains the homogeneous meshwork of the NL, which may be essential for chromatin distribution and nuclear mechanical properties.

## Introduction

The nuclear envelope (NE) is involved in many nuclear activities, including maintenance of the structural integrity of the nucleus, chromatin organization, gene expression, and DNA repair (Burke and Stewart, 2013; Wong et al., 2022). The NE is composed of the outer nuclear membrane, the inner nuclear membrane (INM), nuclear pore complexes (NPCs), and the nuclear lamina (NL), which is a filamentous structure underlying the INM. In the NL, intermediate filament lamins, which are the principal components, and associated proteins, such as nuclear membrane proteins, transcription factors, and chromatin regulators, form a 10–30 nm thick filamentous network (de Leeuw et al., 2018; Wilson and Foisner, 2010). The NL also interacts with heterochromatin domains called lamina-associated domains (LADs; Briand and Collas, 2020; van Steensel and Belmont, 2017) and helps to regulate gene expression and the three-dimensional (3D) organization of the genome (Kim et al., 2019; Rullens and Kind, 2021). This densely packed structure containing proteins and chromatin in the NL defines the mechanical properties of the nucleus and supports nuclear stability and integrity (Kalukula et al., 2022).

Lamins are divided into A- and B-types based on their biochemical properties, expression patterns, and behaviors during mitosis. In mammals, there are three lamin genes, one A-type (*LMNA*) and two B-types (*LMNB1* and *LMNB2*). *LMNA* is alternatively spliced to produce two major isoforms called lamins A and C. The *Drosophila* genome possesses two genes encoding Lamin C and Lamin Dm0, which are designated A- and B-types, respectively (Melcer et al., 2007). Lamin A/C is expressed in differentiated cells and is widely distributed within the nucleoplasm and at the NE. By contrast, B-type lamins are constitutively expressed in all cell types and localize mainly to the nuclear periphery. B-type lamins and prelamin A, but not Lamin C, have a C-terminal CaaX motif in which the cysteine residue is farnesylated. During the processing of prelamin A into mature Lamin A, the C-terminal 15 amino acids, including the farnesylated cysteine, are cleaved by Zmpste24. These posttranslational modifications are important for proper localization and folding of lamins (Dechat et al., 2008).

Lamins consist of an N-terminal head domain, a coiled-coil central rod domain, and a C-terminal Ig-fold domain. This structure is well-conserved among humans, mice, *Caenorhabditis elegans*, *Xenopus laevis*, and *Drosophila* (Stuurman et al., 1998; Tenga and Medalia, 2020). Lamin proteins form a dimer via the coiled-coil rod domain, which further assembles into head-to-tail polymers, which laterally associate to form mature filaments (Ben-Harush et al., 2009). Recent high-resolution structural analysis of mouse embryonic fibroblasts revealed that tetrameric lamin filaments (termed protofilaments) with a thickness of 3.5 nm form a meshwork at the nuclear periphery (Turgay et al., 2017). However, at least in vitro, bacterially expressed

[1]Department of Life Science, College of Science, Rikkyo University, Tokyo, Japan; [2]Department of Molecular Biology, Keio University School of Medicine, Tokyo, Japan; [3]Department of Chromosome Science, National Institute of Genetics, Mishima, Japan; [4]Laboratory for Functional Non-Coding Genomics, RIKEN Center for Integrative Medical Sciences, Yokohama, Japan; [5]Japan Science and Technology Agency (JST), Precursory Research for Embryonic Science and Technology (PRESTO), Saitama, Japan.

Correspondence to Satoshi Goto: stgoto@rikkyo.ac.jp.



lamin protofilaments further assemble into large filaments or paracrystal arrays (Stuurman et al., 1998). This suggests that another factor is necessary for the proper assembly of lamins.

Mutations of lamin genes cause human diseases termed laminopathies, including muscle, metabolic, and neuronal diseases, and accelerated aging (Ho and Hegele, 2019; Worman and Bonne, 2007). Most laminopathies result from dysfunctional variants of *LMNA*, whereas a few have been linked to variants of *LMNB1* and *LMNB2*. Many disease-causing mutant lamins exhibit altered nuclear localization and accumulation (Cowan et al., 2010; Dutta et al., 2018; Hübner et al., 2006). In these nuclei, perturbed nuclear shape, nuclear fragility, disrupted 3D genome organization, and abnormal gene expression have been reported (Bertero et al., 2019; Earle et al., 2020; Paulsen et al., 2017; Shah et al., 2021; Zwerger et al., 2013), suggesting that the proper localization and meshwork of lamins are possibly important for nuclear function. However, this has not been proven. Furthermore, it is unknown which molecule maintains the uniform distribution of the lamin meshwork at the nuclear periphery.

In the present study, we identified PIGB as the molecule responsible for the uniform B-type lamin meshwork and well-organized NL in *Drosophila*. Loss of *PIGB* resulted in irregular aggregation of Lamin Dm0 at the nuclear periphery and the heterogeneous distribution of lamin-binding proteins. To our knowledge, this is the first report of the abnormal distribution of a lamin without a pathogenic mutation of the lamin itself. PIGB was originally identified as an enzyme involved in the synthesis of glycosylphosphatidylinositol (GPI; Takahashi et al., 1996). Specifically, it catalyzes the addition of the third mannose. GPI is synthesized in the endoplasmic reticulum (ER) in mammalian cells; however, we previously reported that *Drosophila* PIGB localizes to the INM via Lamin Dm0 (Yamamoto-Hino et al., 2018, 2020). Although the expression of wild-type (WT) PIGB completely rescues the lethality of the *PIGB* mutant, the expression of ER-localized PIGB does not (Yamamoto-Hino et al., 2018). This suggests that the nuclear localization of PIGB is essential for its function. Here, we showed that the function of PIGB in the NE is independent of its GPI synthesis activity because expression of PIGB lacking mannosyltransferase activity rescued the abnormal distributions of Lamin Dm0 and lamin-binding proteins in the *PIGB* mutant. We investigated how PIGB affects the formation of LADs and nuclear mechanical properties. The *PIGB* mutant had more small LADs in introns and fewer binding sites in centromeres than the WT. We measured nuclear mechanical properties using microneedle-based micromanipulation and found that nuclei were softer in the *PIGB* mutant. Finally, muscle structure was disrupted in the *PIGB* mutant, which was rescued by expressing enzymatically inactive PIGB. These data indicate that PIGB is essential for the organization of the NL and proper nuclear function, in addition to GPI synthesis.

## Results

### PIGB overexpression leads to mislocalization of Lamin Dm0
To elucidate the function of PIGB at the nuclear membrane, we overexpressed myc-tagged WT PIGB (wtPIGBmyc). In WT (Mef2>lacZ) larval wall skeletal muscle, endogenous PIGB

colocalized with Lamin Dm0 in the NE as we previously reported (Fig. 1 A, left panel; Yamamoto-Hino et al., 2018). However, in larvae overexpressing PIGBmyc (Mef2>wtPIGBmyc), PIGB localized to regions outside the NE and most did not colocalize with Lamin Dm0 (Fig. 1 A, right panel). At high magnification, overexpressed wtPIGBmyc was also observed in the nucleoplasm, where PIGB was not natively localized in the WT (Fig. 1 B, arrowheads). wtPIGBmyc in the nucleoplasm was associated with Lamin Dm0. To clarify whether wtPIGBmyc and Lamin Dm0 in the nucleoplasm were connected to the INM, we performed 3D observations. wtPIGBmyc and Lamin Dm0 in the nucleoplasm were continuous with, not independent of, the nuclear membrane, indicating that wtPIGBmyc overexpression results in dramatic membrane deformation including nuclear invaginations from the INM. In addition, wtPIGBmyc was also distributed in the cytoplasmic region around the NE (Fig. 1, B and C). Simultaneous observation revealed that wtPIGBmyc colocalized with Calr-GFP, an ER marker (Fig. 1 D), indicating that cytoplasmic wtPIGBmyc localizes to the ER. This is a common phenomenon when integral NE proteins are overexpressed, probably due to saturation of the binding site in the NE (Wilkie et al., 2011). Interestingly, Lamin Dm0 also colocalized with wtPIGBmyc in the ER (Fig. 1, B and C, arrows). Taken together, these results demonstrated that PIGB in the NE strongly interacts with Lamin Dm0 and thereby localizes it to the NL.

### PIGB is essential for the formation and maintenance of the well-organized NL
Next, we observed the localization of Lamin Dm0 in the body wall muscle of *PIGB*-deficient (*PIGB*[13]) larvae, which harbor a 938 bp deletion in the *PIGB* gene that removes the first ATG by imprecise excision (Yamamoto-Hino et al., 2018). Nuclei in body wall muscle are flat and thick, allowing observation of a wide field of view just below the nuclear membrane (Fig. S1 A). In the WT *PIGB*[27], in which the inserted transposon was properly excised and which was generated simultaneously with *PIGB*[13], Lamin Dm0 was homogenously distributed underneath the INM (Fig. 2 A, left panel). However, in *PIGB*[13], the homogeneous distribution of Lamin Dm0 was impaired (Fig. 2 A, right panel). We quantified the heterogeneity in the distribution of Lamin Dm0 as variations in the signal intensity of Lamin Dm0. We defined the distribution index (D.I.) as the value obtained by normalizing the intensity variation by the mean fluorescence intensity of nuclear images. We captured images of 10–12 nuclei per individual, calculated the D.I., and compared the mean D.I. among six individuals (D.I. was 0.230 ± 0.022 versus 0.408 ± 0.044, P = 4.91E–06 by the unpaired two-tailed *t* test, Fig. 2 B). To confirm that the phenotype is attributable to the *PIGB* deletion, we created another null mutant using CRISPR editing. This mutant, known as *PIGB*[CRP5], carries a 5-bp deletion spanning from nucleotide 74 to nucleotide 78 relative to the first ATG of the *PIGB* gene, resulting in a protein of only 27 amino acids. Similar to *PIGB*[13], this mutant larva exhibited mortality during the late larval stage. *PIGB*[CRP2], which was generated concurrently, does not possess the mutation and served as the WT control. We verified the absence of PIGB protein expression in the larval muscle of *PIGB*[CRP5] using Western blotting (Fig. S1 B).

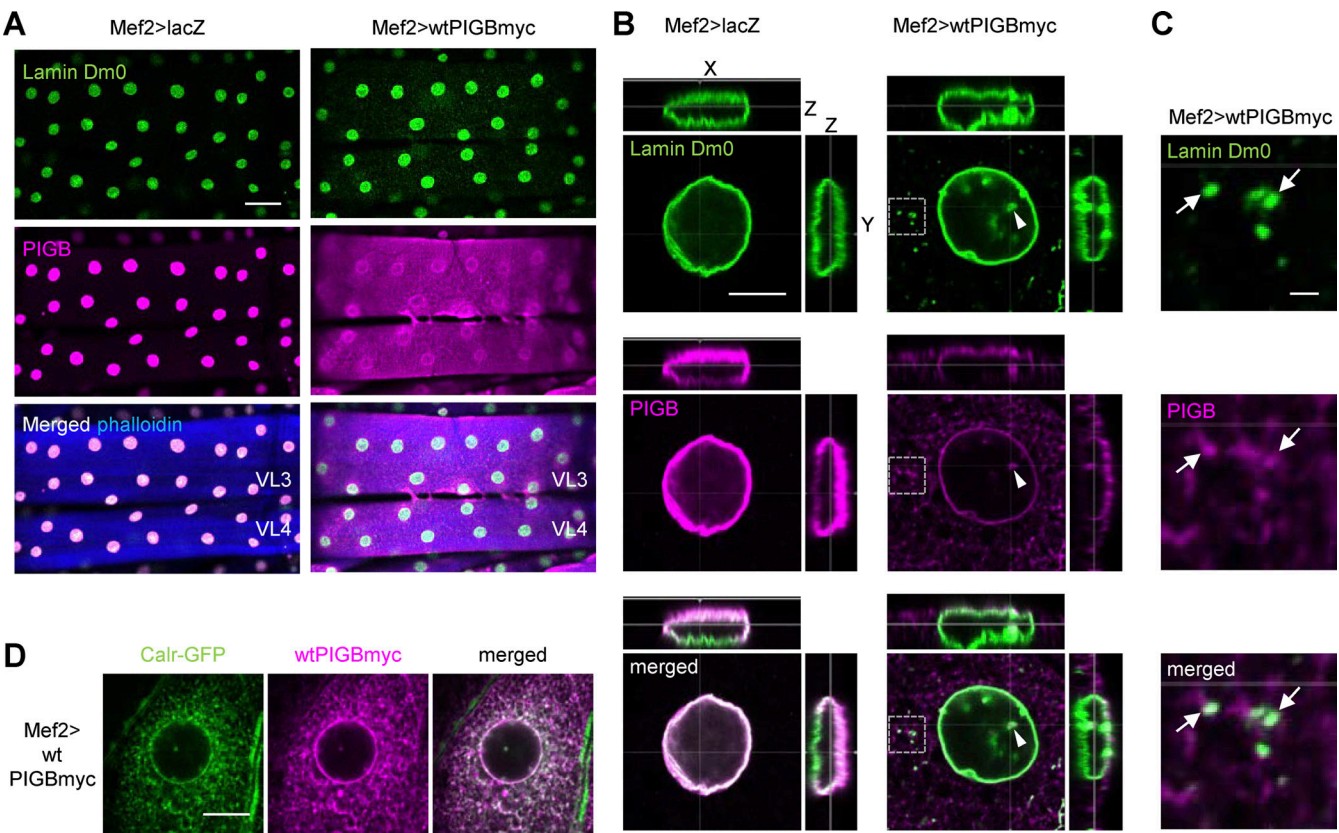

Figure 1. **PIGB overexpression leads to mislocalization of Lamin Dm0. (A and B)** WT (Mef2>lacZ) and WT PIGB-overexpressing (Mef2>wtPIGBmyc) larval wall skeletal muscle (VL3 and VL4) stained for Lamin Dm0 (green) and PIGB (magenta) at low magnification (A; bar, 50 μm) and 3D high magnification (B; bar, 10 μm). White arrowheads indicate colocalization of Lamin Dm0 and PIGB in the nucleoplasm. Actin was stained with phalloidin (blue). **(C)** Enlarged view of the area enclosed by the dotted box in B. White arrows indicate colocalization of Lamin Dm0 and PIGB in the cytoplasm. Bar, 1 μm. **(D)** Colocalization of Calr-GFP (ER marker, green, stained with an anti-GFP antibody) and overexpressed wtPIGBmyc (magenta, stained with an anti-PIGB antibody) using Mef2-Gal4. Bar, 10 μm.

Nuclei of $PIGB^{CRP5}$, similar to those of $PIGB^{13}$, exhibited a heterogeneous distribution of Lamin Dm0 (Fig. S1, C and D, D.I. was 0.249 ± 0.030 versus 0.444 ± 0.030, P = 6.01E–07 by the unpaired two-tailed $t$ test). Moreover, NPCs were regularly arranged along the lamin network of the NE in $PIGB^{27}$, whereas loss of $PIGB$ caused NPC clustering (Fig. 2, A and B; D.I. was 0.243 ± 0.036 versus 0.423 ± 0.017, P = 6.93E–07 by the unpaired two-tailed $t$ test). The results were similar in $PIGB^{CRP5}$ (Fig. S1, C and D; D.I. was 0.255 ± 0.011 versus 0.457 ± 0.025, P = 5.51E–09 by the unpaired two-tailed $t$ test). This pattern of NPCs is similar to that observed in $Lamin\ Dm0$ mutants reported previously (Osouda et al., 2005) and is shown in Fig. S1 E. The $Lam^{K2}$ allele, which is characterized by the additional nucleotide sequence "CTGC" between G460 and A461 as well as a frameshift mutation after amino acid 153, was employed as a loss-of-function mutant for $Lamin\ Dm0$. Western blotting of larval muscle tissue from the $Lam^{K2}$ mutant revealed that Lamin Dm0 protein expression was below the detection threshold (Fig. S1 F). GFP-tagged Nup107, a component of the NPC, localized complementarily to Lamin Dm0 in $PIGB^{13}$ (Fig. S1 G). This suggests that NPCs are separate from the Lamin Dm0 meshwork. A- and B-type lamins are organized into distinct networks at the nuclear periphery (Shimi et al., 2015). The localization of the $Drosophila$ A/C-type lamin,

Lamin C, appeared to be normal in $PIGB^{13}$ (Fig. 2 A). However, the mutant exhibited slightly greater heterogeneity (Fig. 2 B; D.I. was 0.200 ± 0.009 versus 0.225 ± 0.023, P = 0.032 by the unpaired two-tailed $t$ test). A similar result was observed when comparing $PIGB^{CRP2}$ and $PIGB^{CRP5}$ (Fig. S1, C and D; D.I. was 0.211 ± 0.013 versus 0.245 ± 0.021, P = 0.007 by the unpaired two-tailed $t$ test). In addition, 3D observation showed that 48.7% ($n$ = 117) of nuclei in the larval body wall muscle of $PIGB^{13}$ formed ectopic lamina, including Lamin C, in the nucleoplasm compared with 2.0% ($n$ = 149) of nuclei in $PIGB^{27}$ (Fig. 2 C, right panel, arrow). The distribution of Lamin C, although less affected than that of Lamin Dm0, seemed to be influenced by the absence of PIGB. We also observed the distribution of the lamin-binding protein Otefin (Ote; Ashery-Padan et al., 1997), a $Drosophila$ LEM domain protein (Barton et al., 2014). Despite the weak signals and high background noise in its staining, Ote was homogenously distributed in the INM in $PIGB^{27}$, but not in $PIGB^{13}$, similar to Lamin Dm0 (Fig. 2, A and B; D.I. was 0.215 ± 0.011 versus 0.256 ± 0.024, P = 0.0214 by the unpaired two-tailed $t$ test).

We also investigated the distribution of lamin in cells other than muscle cells. Many cells do not exhibit flat nuclei like muscle cells and thus it can be challenging to observe the nuclear

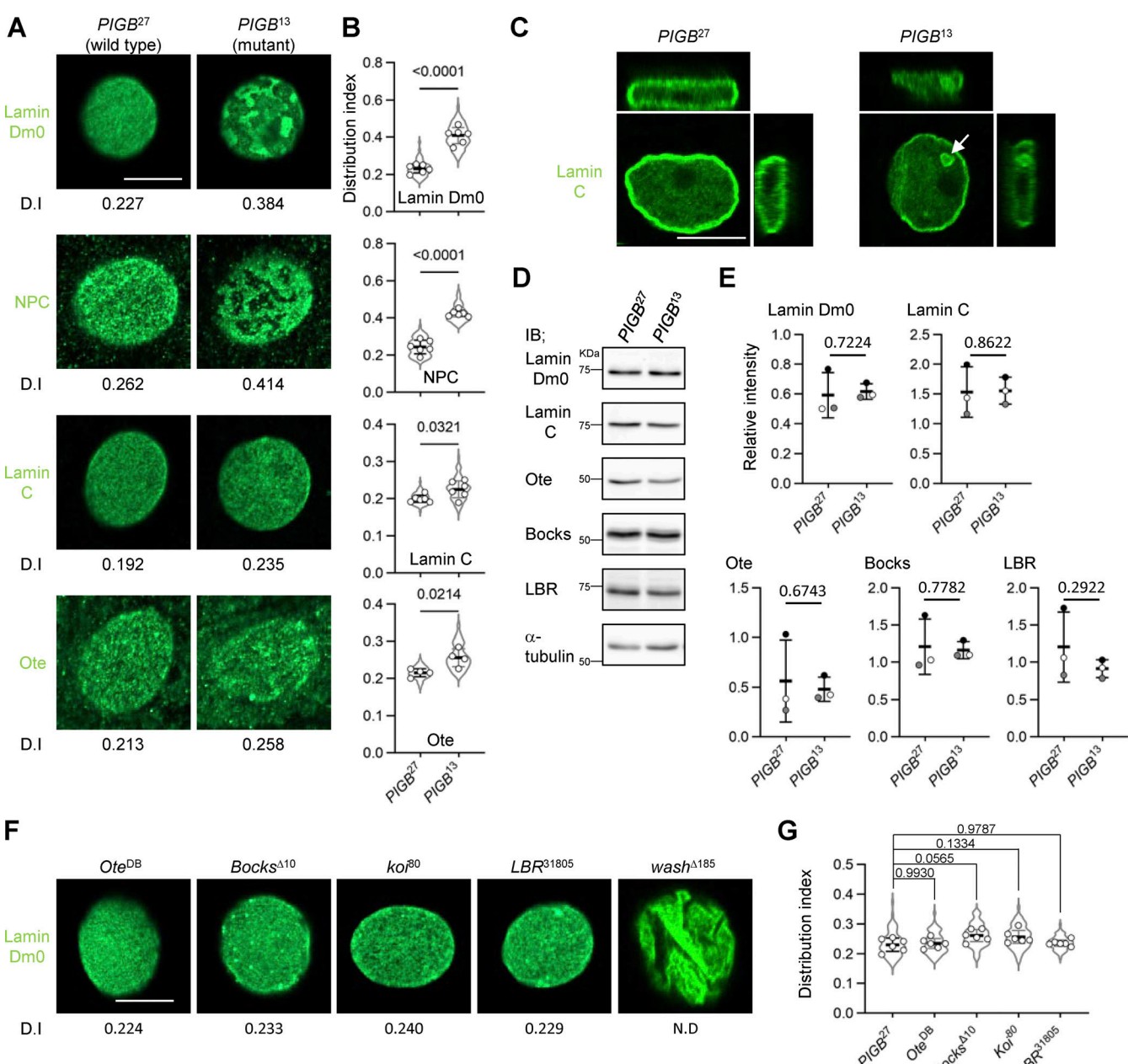

Figure 2. **Loss of PIGB leads to disorganization of the NL. (A)** Distributions of Lamin Dm0, NPCs, Lamin C, and Ote in a nucleus of larval wall skeletal muscle in *PIGB*[27] (WT) and *PIGB*[13] (mutant). The number at the bottom of the image is the D.I., which was obtained by normalizing the SD of intensity by the mean intensity value. Bar, 10 µm. **(B)** Quantification of the distributions of Lamin Dm0, NPCs, Lamin C, and Ote in a nucleus of larval wall skeletal muscle in the *PIGB*[27] and *PIGB*[13] larvae shown in A. 10–12 nuclei per individual were measured and the average value was plotted for six individuals (white circle). The thick black horizontal bar and thin gray horizontal bar show the mean and SD of six biological replicates, respectively; >60 nuclei analyzed per strain. The superimposed violin plot shows the distribution of the D.I. The number at the top of the graph is the P value (biological replicates = 6) calculated using the unpaired two-tailed *t* test. **(C)** 3D view of a nucleus stained with an anti-Lamin C antibody in *PIGB*[27] and *PIGB*[13]. The white arrow indicates the ectopic lamina in the nucleoplasm of *PIGB*[13]. Bar, 10 µm. **(D)** Immunoblot analysis of Lamin Dm0, Lamin C, Ote, Bocks, and LBR in larval carcasses of *PIGB*[27] and *PIGB*[13]. α-Tubulin was used as a loading control. **(E)** Quantification of the immunoblot analyses is shown in D. Lysates from 10 carcasses of *PIGB*[27] and *PIGB*[13] larvae were subjected to three independent experiments. The intensity of each band was normalized against that of α-tubulin. Normalized values in *PIGB*[13] were compared with those in *PIGB*[27] as a control. The thick black horizontal bar and thin gray horizontal bar show the mean and SD, respectively. White, gray, and black circles correspond to each pair in the three experiments. The number at the top of the graph is the P value (experimental replicates = 3) calculated using the paired two-tailed *t* test. **(F)** Distributions of Lamin Dm0 in larval wall skeletal muscle in mutants of the lamin-binding proteins Ote, Bocks, Koi, LBR, and wash. The number at the bottom of the image is the D.I. Bar, 10 µm. **(G)** Quantification of the distributions of Lamin Dm0 in mutants of the lamin-binding proteins shown in F. 10–12 nuclei per individual were measured and the average value was plotted for six individuals (white circle). The thick black horizontal bar and thin gray horizontal bar show the mean and SD of six biological replicates, respectively; >60 nuclei analyzed per strain. The superimposed violin plot shows the distribution of the D.I. The number at the top of the graph is the P value versus *PIGB*[27] (biological replicates = 6) calculated using a one-way ANOVA with Tukey's multiple comparison test. Source data are available for this figure: SourceData F2.

membrane surface. Consequently, we performed observations by cutting through the central plane of the nucleus using a confocal microscope (Fig. S1 H). When lamin was observed in WT (*PIGB*[27]) muscle cells at the central nucleus plane, a ring-shaped signal was consistently observed. In PIGB mutants, the lamin distribution appeared heterogeneous, resulting in distinct regions with varying signal intensities, unlike in the WT. However, in cells of the wing disc, ventral nerve cord neurons, and fat body cells, the lamin distribution was unaltered compared with the WT (Fig. S1 I). Taken together, our findings suggest that PIGB plays a crucial role in the formation and maintenance of a well-organized NL in muscle cells.

Next, we investigated whether *PIGB* depletion affects the expression levels of nuclear proteins. Immunoblotting of Lamin Dm0 and Lamin C in larval carcasses containing muscles showed that the levels of these proteins were not changed in *PIGB*[13]. In addition, the expression levels of Ote, Bocks (Wagner et al., 2004a), another LEM domain protein, and Lamin B receptor (LBR; Wagner et al., 2004b) were similar in *PIGB*[27] and *PIGB*[13] (Fig. 2, D and E). These results indicate that PIGB does not affect the expression levels of proteins located in the NL. Furthermore, Lamin Dm0 undergoes various posttranslational modifications (Murray-Nerger and Cristea, 2021). However, despite extended exposure in immunoblot analysis (Fig. S1 J), no discernible difference in band patterns between *PIGB*[27] and *PIGB*[13] was observed. This suggests there are no detectable variations in posttranslational modifications of Lamin Dm0 between *PIGB*[27] and *PIGB*[13] as determined by immunoblotting.

There are many lamin-binding proteins in the nuclear membrane. Therefore, we investigated whether other lamin-binding proteins besides PIGB affect the distribution of Lamin Dm0. *Drosophila* expresses three LEM domain proteins: Ote, Bocks, and dMAN (Barton et al., 2014; Wagner et al., 2006). We could not obtain *dMAN1* mutant homozygous larvae. The other two mutants had normal distributions of Lamin Dm0 (Fig. 2, F and G; D.I. values of *Ote*[DB] and *Bocks*[Δ10] were 0.235 ± 0.017 and 0.261 ± 0.020, respectively). Klaroid (Koi) is a SUN domain protein that interacts with microtubules in the cytoplasm through a KASH domain protein (Kracklauer et al., 2007). In the *Koi* mutant, Lamin Dm0 was well-organized (Fig. 2, F and G; D.I. was 0.257 ± 0.023). LBR has multiple transmembrane segments, similar to PIGB. There is no *LBR* mutant; therefore, we obtained an *LBR* null mutant by replacing the LBR locus with ey-DsRed using the CRISPR/Cas method. We confirmed that LBR protein was depleted in larval body wall muscle by immunoblotting (Fig. S1 K). However, the distribution of Lamin Dm0 was not disturbed (Fig. 2, F and G; D.I. was 0.236 ± 0.010). The D.I. of no mutant significantly differed compared with the WT *PIGB*[27], as determined by a one-way ANOVA with Tukey's multiple comparison test (Fig. 2 G). Wash is a member of the Wiskott-Aldrich syndrome protein family and functions in the nucleus to modulate nuclear organization (Verboon et al., 2015). Its deletion causes an abnormal wrinkle-like morphology of nuclei; therefore, the distribution of Lamin Dm0 appeared to be heterogeneous (Fig. 2 F). However, 3D observation showed that Lamin Dm0 was uniformly distributed along the nuclear membrane (Fig. S1 L). Consequently, no D.I. calculations were performed.

These data indicate that PIGB is a unique protein essential for the formation and maintenance of the well-organized NL.

In addition, phalloidin-stained fibers were observed in nuclei of *PIGB*[13], suggesting that actin filaments form in the nucleus. Almost all WT third instar wandering larvae became pupae within 12 h, while *PIGB*-deficient wandering larvae existed for about 5 d and then became pupae or gradually died. *PIGB*[27] and *PIGB*[13] larvae within 12 h of the wandering stage (*PIGB*[27] [<12 h] and *PIGB*[13] [<12 h]) rarely exhibited actin fibers in the nuclei (*n* = 12, Fig. 3, A and B). However, 2 d later, 58% of *PIGB*-deficient larvae (*PIGB*[13] [>2 d]; *n* = 12) showed actin fibers in nuclei (Fig. 3, A and B). A similar result was obtained using another mutant, *PIGB*[CRP2], and *PIGB*[CRP5] (0% of *PIGB*[CRP2] [<12 h], 0% of *PIGB*[CRP5] [<12 h], and 41.6% of *PIGB*[CRP5] [>2 d], *n* = 12; Fig. 3 B). At high magnification, intranuclear actin fibers localized complementarily to DAPI staining in *PIGB*[13] [>2 d] (Fig. 3 C). To confirm the presence of actin fibers in nuclei of *PIGB*[13] [>2 d], we performed 3D observations by labeling the nuclear membrane with Lamin C. We observed phalloidin signals in nuclei, indicating that actin, distinct from muscle actin fibers, localized to the nuclear interior (Fig. 3 D). Although monomeric G-actin and polymeric F-actin are present in the nucleus, they are not detected by phalloidin (Kristó et al., 2016). Therefore, no such fibers were observed in the muscle nuclei of WT larvae. Human Lamins A and B bind directly to purified actin in vitro (Simon et al., 2010), and *Lamin C* null mutant and N-terminal truncated Lamin C-expressing *Drosophila* larvae display actin polymers in nuclei (Dialynas et al., 2010; Schulze et al., 2009). Furthermore, in mammals, it has been reported that Emerin, a LEM-domain protein, directly binds to nuclear actin, regulating processes such as mRNA transcription and chromatin remodeling (Wilson et al., 2005). These findings imply that loss of PIGB would impact potential functions of Lamin C and LEM-domain proteins, including Ote, in the regulation of nuclear actin polymerization and formation or stabilization of a cortical actin network in the NL.

We observed the structure of the nuclear membrane by transmission electron microscopy. Compared with *PIGB*[27] [<12 h] nuclei, the perinuclear space in the nuclei of *PIGB*[13] [<12 h] was wider (thin white arrows in Fig. 3, E and F; 27.0 ± 3.8 nm versus 39.1 ± 3.5 nm, P = 0.0002 by a one-way ANOVA with Tukey's multiple comparison test). Such separation of the inner and outer nuclear membranes was reported in imaginal discs of the *Lamin C* null mutant (Schulze et al., 2009). On the other hand, the perinuclear space was narrower in the nuclei of *PIGB*[13] [>2 d] (Fig. 3, E and F; 16.1 ± 2.4 nm, P = 0.0006 versus *PIGB*[27] [<12 h] by a one-way ANOVA with Tukey's multiple comparison test). The cytoplasmic side of the nucleus was covered with an ~10 nm-thick dense layer (black arrows in Fig. 3 E). In addition, irregularly shaped dense structures were observed in the nucleoplasm of *PIGB*[13] [>2 d] larvae (thick white arrows in Fig. 3 E).

## GPI synthesis activity of PIGB is not required for organization of the NL

PIGB was originally identified as an enzyme involved in the synthesis of GPI. Specifically, it catalyzes the addition of the third mannose. We investigated whether mannosyltransferase

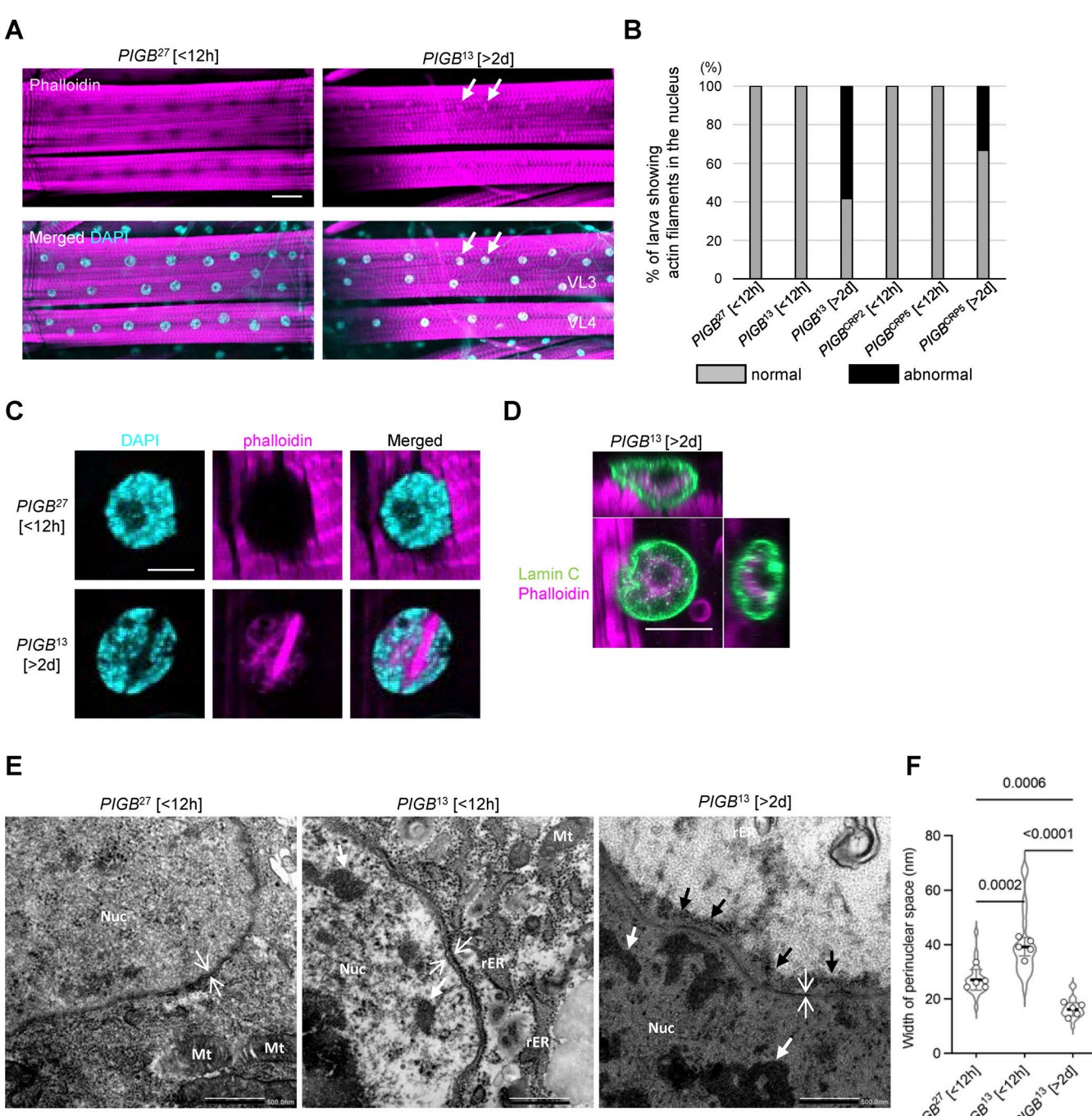

Figure 3. **Abnormal localization of actin fibers in the nucleoplasm and defects of the NE. (A)** Phalloidin staining (magenta) of larval wall skeletal muscle in $PIGB^{27}$ within 12 h of the wandering stage ($PIGB^{27}$ [<12 h]) and $PIGB^{13}$ 2 d later ($PIGB^{13}$ [>2 d]) at low magnification. Arrows indicate actin polymerization in nuclei. Nuclei were stained with DAPI (cyan). Bar, 50 μm. **(B)** Percentage of larvae with nuclear actin in wall skeletal muscle in $PIGB^{27}$ [<12 h], $PIGB^{13}$ [<12 h], $PIGB^{13}$ [>2 d], $PIGB^{CRP2}$ [<12 h], $PIGB^{CRP5}$ [<12 h], and $PIGB^{CRP5}$ [>2 d]. 12 individuals per strain were observed. Normal (gray): individuals with no actin in the nucleus. Abnormal (black): individuals with actin in the nucleus. **(C)** Nuclei in A at high magnification. Bar, 10 μm. **(D)** 3D observation of actin (magenta) in $PIGB^{13}$[>2 d]. The NE was labeled with Lamin C (green). Actin filaments localized to the nucleoplasm surrounded by Lamin C. Bar, 10 μm. **(E)** Nuclear membrane observed by transmission electron microscopy. White thin arrows indicate the width of the perinuclear space. White thick arrows indicate typical examples of irregularly shaped dense structures observed in the nucleoplasm of $PIGB^{13}$ [<12 h] and $PIGB^{13}$ [>2 d] larvae. Black arrows indicate the dense layer around nuclei in $PIGB^{13}$ [>2 d]) larvae. Nuc; nucleus, rER; rough endoplasmic reticulum, Mt; mitochondria. Bar, 500 nm. **(F)** Dot blot of the width of the perinuclear space in $PIGB^{27}$ [<12 h], $PIGB^{13}$ [<12 h], and $PIGB^{13}$ [>2 d] larvae. The thickness was measured at four to five locations per nucleus and the average value was plotted for five images (white circle). The thick black horizontal bar and thin gray horizontal bar show the mean and SD of five biological replicates, respectively. The superimposed violin plot shows the distribution of the thickness of the perinuclear space ($n$ = 25 for $PIGB^{27}$ [<12 h], $n$ = 24 for $PIGB^{13}$ [<12 h], and $n$ = 23 for $PIGB^{13}$ [>2 d]). The number at the top of the graph is the P value (biological replicates = 5) calculated using a one-way ANOVA with Tukey's multiple comparison test.

activity of PIGB is required for the organization of the NL. A membrane topology model of eukaryotic mannosyltransferases suggests that PIGB has 11 transmembrane helices (Fig. S2 A). Multiple alignments of eukaryotic mannosyltransferases revealed a few conserved amino residues located in the luminal loop (Albuquerque-Wendt et al., 2019). By mutating these amino acids to alanine, we attempted to generate a *PIGB* mutant that lacked mannosyltransferase activity. We assayed PIGB mannosyltransferase activity by testing whether cell-surface expression of a GPI-anchored protein was restored in *PIGB*-deficient CHO cells (class B cells). Surface expression of urokinase plasminogen receptor (uPAR) in these cells was fully restored by expression of the wtPIGBmyc (Fig. 4 A, red dotted line), but not by expression of PIGBmyc with D28A and E29A mutations (ΔactPIGBmyc; Fig. 4 A, gray line). Immunoblot analysis showed that the expression level of ΔactPIGBmyc was slightly lower than that of wtPIGBmyc (Fig. 4 B, left panel, 1.0 ± 0.04 versus 0.80 ± 0.05, P = 0.006 by the unpaired two-tailed *t* test). Next, we transfected *Drosophila* S2 cells with ΔactPIGBmyc to check their localization and expression level. ΔactPIGBmyc localized to the nuclear membrane identical with wtPIGBmyc in these cells (Fig. 4 C), although the expression level of ΔactPIGBmyc was lower than that of wtPIGBmyc (Fig. 4 B, right panel, 1.0 ± 0.164 versus 0.11 ± 0.009, P = 0.0007 by the unpaired two-tailed *t* test).

We examined whether ΔactPIGBmyc restores the heterogeneous distribution of Lamin Dm0 in the larval wall muscle of *PIGB*[13]. wtPIGBmyc and PIGBmyc variants were expressed using the Gal4-UAS system. This system involves crossing the Mef2-Gal4 driver, which expresses the Gal4 transcription factor in a muscle-specific pattern, to a responder possessing a transgene driven by a UAS element. The UAS element is bound by Gal4, which results in the activation of transgene expression. A higher number of UAS elements is expected to increase expression. The effects of expression of the responder gene in a muscle-specific pattern are assessed in the resulting progeny. We generated a fly strain carrying 3UAS-ΔactPIGBmyc inserted into chromosome 68A4, which is in the same position as the insertion site of wtPIGBmyc. The number in front of "UAS" such as "3UAS" indicates the number of consecutively placed UAS elements. When 3UAS-ΔactPIGBmyc (68A4) was expressed in *PIGB*[13] muscles using Mef2-Gal4, Lamin Dm0 had a completely homogenous distribution identical to the effect of wtPIGBmyc expression (Fig. 4 D, upper panel and Fig. 4 E, D.I. was 0.237 ± 0.014 for Mef2 control, 0.463 ± 0.048 for Mef2, *PIGB*[13], 0.264 ± 0.029 for 3UAS-wtPIGBmyc (68A4), and 0.262 ± 0.044 for 3UAS-ΔactPIGBmyc (68A4); see Fig. S2 B for the results of statistical tests), although ΔactPIGBmyc expression was very low (Fig. 4, F and G). The expression level of wtPIGBmyc was 18-fold higher than that of endogenous PIGB, while the expression level of 3UAS-ΔactPIGBmyc (68A4) was only 2.59 ± 0.55% (Fig. 4 G). To avoid the influence of the genetic background on the distribution of Lamin Dm0 and to enhance expression, we generated a new fly strain carrying 20UAS-ΔactPIGBmyc inserted into chromosome 55C4. The expression level of 20UAS-ΔactPIGBmyc (55C4) improved to 25.3 ± 5.4% of endogenous PIGB (Fig. 4, F and G) and the distribution of Lamin Dm0 was rescued (Fig. 4, D and

E, D.I. was 0.256 ± 0.017, P = 0.949 versus Mef2 and P < 0.0001 versus Mef2, *PIGB*[13] by a one-way ANOVA with Tukey's multiple comparison test). We also examined whether ER-localized PIGB (ERPIGB), which possesses mannosyltransferase activity but does not localize to the NE, was unable to restore the heterogeneous distribution of Lamin Dm0 in *PIGB*[13].

The expression level of 3UAS-ERPIGBmyc (68A4) was low (15.1 ± 0.70% of endogenous PIGB) but higher than that of 3UAS-ΔactPIGBmyc (68A4; Fig. 4, F and G). Expression of ERPIGBmyc did not rescue the heterogeneous distribution of Lamin Dm0 in *PIGB*[13] (Fig. 4, D and E, D.I. was 0.512 ± 0.060, P < 0.0001 versus Mef2 and P = 0.2727 versus Mef2, *PIGB*[13] by a one-way ANOVA with Tukey's multiple comparison test). Taken together, these results demonstrate that the localization of PIGB to the NE is crucial for the uniform distribution of Lamin Dm0 and that GPI synthesis activity is not required. Consistently, expression of 3UAS-wtPIGBmyc (68A4), 3UAS-ΔactPIGBmyc (68A4), and 20UAS-ΔactPIGBmyc (55C4) restored the clustering of NPCs in *PIGB*[13], while 3UAS-ERPIGBmyc (68A4) did not (Fig. 4, D and E, lower panel, D.I. was 0.211 ± 0.016 for Mef2 control, 0.466 ± 0.023 for Mef2, *PIGB*[13], 0.214 ± 0.011 for 3UAS-wtPIGBmyc (68A4), 0.201 ± 0.012 for 3UAS-ΔPIGBmyc (68A4), 0.199 ± 0.012 for 20UAS-ΔPIGBmyc (55C4), and 0.444 ± 0.035 for 3UAS-ERPIGBmyc (68A4); see Fig. S2 B for the results of statistical tests). In addition, nuclear actin fibers were decreased when ΔactPIGBmyc was expressed in *PIGB*[13]. The percentage of larvae with nuclear actin fibers decreased from 47.2% (n = 36) in *PIGB*[13] [>2 d] larvae to 11.1% (n = 36) in *PIGB*[13] [>2 d] larvae expressing 3UAS-ΔactPIGBmyc (68A4) and 20UAS-ΔactPIGBmyc (55C4; Fig. 4 H). Expression of ΔactPIGBmyc did not rescue the delayed development of third instar *PIGB*[13] larvae; therefore, the reduction of nuclear actin polymerization was not due to restoration of pupal development timing. When ERPIGBmyc was expressed, nuclear actin also disappeared. However, the expression of ERPIGBmyc led to normal pupal development timing and thus it was impossible to sample older larvae; therefore, it remained unclear whether rescue had occurred.

These results indicate that the function of PIGB in the nuclear membrane is independent of its GPI synthesis activity and that only a small amount of PIGB is required for this function.

### Loss of PIGB affects the formation of LADs

Lamins are important for the generation of LADs because they bind to DNA and chromatin (Manzo et al., 2022). It is of great interest to reveal whether the accumulation of Lamin Dm0 in *PIGB*[13] affects the spatial distribution of endogenous chromatin. To investigate this, we performed muscle-specific genome-wide DamID-lamin (Dam-Lam), a method in which muscle-specific expression of Lamin Dm0 fused to *Escherichia coli* DNA adenine methyltransferase (Dam) enables mapping of chromatin at the INM of muscle nuclei (Pindyurin et al., 2016). We expressed Dam-Lam in a muscle-specific manner using a FLP-FRT system. This resulted in the preferential methylation of adenines in GATC motifs of chromatin adjacent to the INM in muscle cells only. Using G[me]ATC-specific restriction enzymes and ligation-mediated amplification, the obtained DNA fragments were sequenced. The generated signals were then normalized to the level of

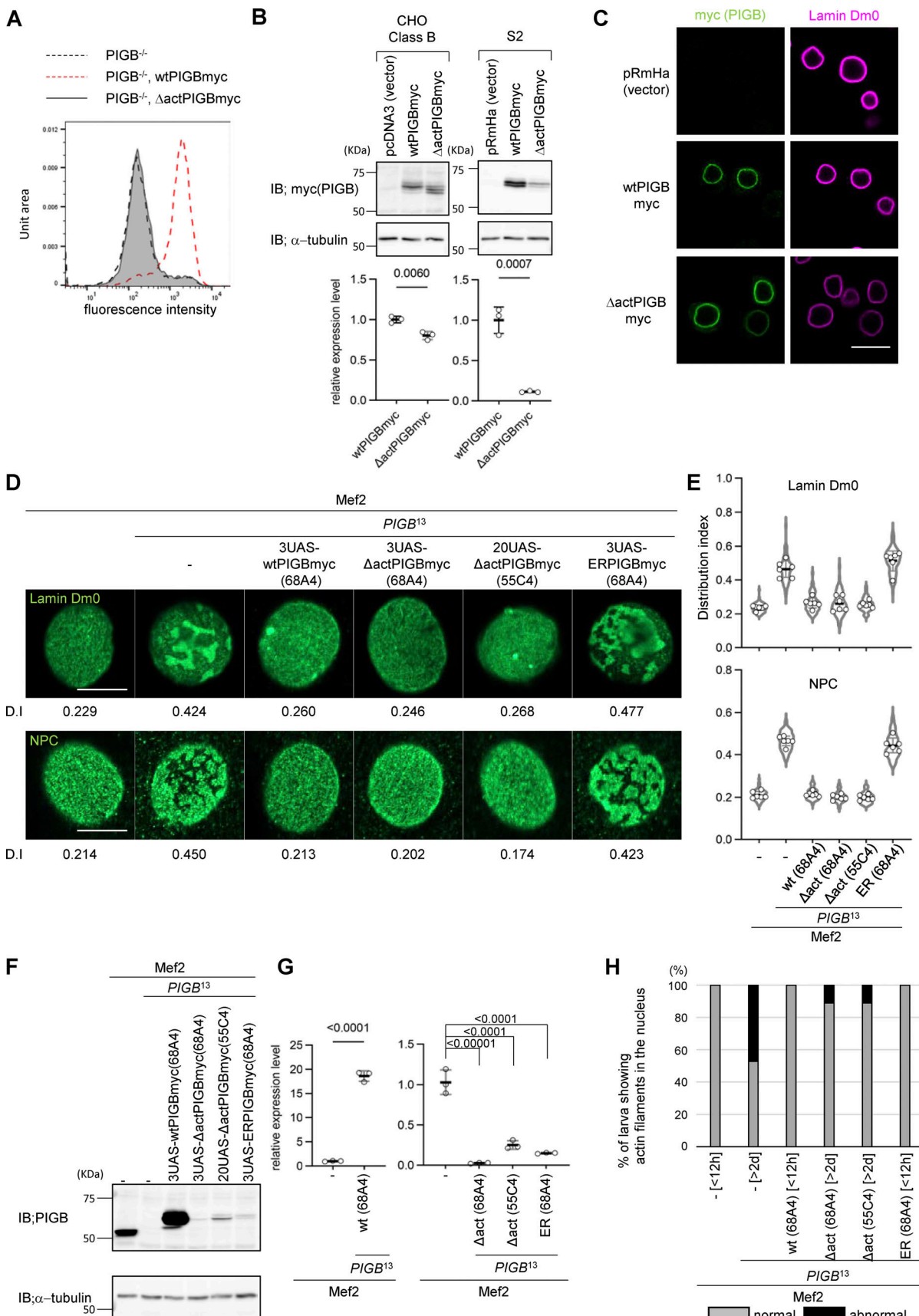

Figure 4. **The GPI synthesis activity of PIGB is not required for the organization of the NL. (A)** Surface expression of uPAR in class B mutant CHO cells transfected with wtPIGBmyc and ΔactPIGBmyc cDNA determined by flow cytometry. The PIGBmyc expression plasmid was transiently co-transfected with the

EGFP expression plasmid (molecular ratio was 4:1). A total of $1 \times 10^4$ GFP-positive cells, which were considered to be transfected with PIGBmyc, were evaluated. Black dotted line, class B mutant cells; red dotted line, class B mutant cells expressing wtPIGBmyc; gray line, class B mutant cells expressing ΔactPIGBmyc. **(B)** Immunoblot analysis of CHO class B (left) and S2 (right) cells expressing wtPIGBmyc and ΔactPIGBmyc. Proteins were detected with anti-myc and anti-α–tubulin (loading control) antibodies. The bottom graph shows the quantification of expressed PIGBmyc. The transfection experiment was performed three times (biological replicates = 3). For each transfection, the immunoblot experiment was performed three times (experimental replicates = 3). From the three experiments, the amounts of α-tubulin and PIGBmyc expressed were calculated, and the amount of PIGBmyc normalized by the amount of α-tubulin was averaged. The average value normalized by wtPIGBmyc as a control was plotted for each transfection (white circle). The thick black horizontal bar and thin gray horizontal bar show the mean and SD of three biological replicates, respectively. The number at the top of the graph is the P value (biological replicates = 3) calculated using the unpaired two-tailed $t$ test. **(C)** Immunofluorescence analysis of S2 cells expressing wtPIGBmyc and ΔactPIGBmyc. Cells were stained with anti-myc (green) and anti-Lamin Dm0 (magenta) antibodies. Bar, 10 μm. **(D)** Rescue of the heterogeneous distributions of Lamin Dm0 (upper panel) and NPCs (lower panel) upon expression of wtPIGBmyc and PIGBmyc variants using Mef2-Gal4. The numbers and letters in parentheses indicate the position of the chromosome in which the transgene was inserted. The number at the bottom of the image is the D.I. Bar, 10 μm. **(E)** Quantification of the distributions of Lamin Dm0 and NPCs in a nucleus of larval wall skeletal muscle upon expression of wtPIGBmyc and PIGBmyc variants using Mef2-Gal4 shown in D. 10–12 nuclei per individual were measured and the average value was plotted for six individuals (white circle). The thick black horizontal bar and thin gray horizontal bar show the mean and SD of six biological replicates, respectively; >60 nuclei analyzed per strain. The violin plot shows the distribution of the D.I. P values (biological replicates = 3) calculated using a one-way ANOVA with Tukey's multiple comparison test are shown in Fig. S2 B. **(F)** Immunoblot analysis of larval body wall muscle of Mef2-Gal4 only and *PIGB*[13] expressing wtPIGBmyc and PIGBmyc variants using Mef2-Gal4. Proteins were detected with anti-PIGB and anti-α–tubulin (loading control) antibodies. **(G)** Quantification of immunoblot analyses shown in F. 3 lysates from 10 carcasses of (Mef2), (Mef2, *PIGB*[13]), and (Mef2, *PIGB*[13] expressing wtPIGBmyc and PIGBmyc variants) larvae were prepared (biological replicates = 3). These lysates were subjected to the immunoblot experiment three times (experimental replicates = 3). From the three experiments, the amounts of α-tubulin and PIGBmyc expressed were calculated, and the amount of PIGBmyc normalized by the amount of α-tubulin was averaged. The average value was plotted for each biological replicate (white circle) normalized by Mef2 as a control. The thick black horizontal bar and thin gray horizontal bar show the mean and SD of three biological replicates, respectively. The number at the top of the graph is the P value (biological replicates = 3) calculated using the unpaired two-tailed $t$ test for (Mef2) versus (Mef2, *PIGB*[13] expressing 3UAS-wtPIGBmyc [68A4]) and a one-way ANOVA with Tukey's multiple comparison test for (Mef2) versus ([Mef2, *PIGB*[13] expressing 3UAS-ΔactPIGBmyc [68A4], 20UAS-ΔactPIGBmyc[55C4], and 3UAS-ERPIGBmyc [68A4]]). **(H)** Percentage of larvae with nuclear actin in wall skeletal muscle for Mef2-Gal4 only (Mef2, –, [<12 h]), Mef2-Gal4, *PIGB*[13] (Mef2, *PIGB*[13] -, [>2 d]), and *PIGB*[13] expressing wtPIGBmyc (Mef2, *PIGB*[13], wt (68A4) [<12 h]), 3UAS-ΔactPIGBmyc (68A4) (Mef2, *PIGB*[13], Δact (68A4) [>2 d]), 20UAS-ΔactPIGBmyc(55C4) ([Mef2, *PIGB*[13], Δact [55C4] [>2 d]), and 3UAS-ERPIGBmyc (68A4) ([Mef2, *PIGB*[13], ER [68A4] [<12 h]) using Mef2-Gal4. 36 individuals per strain were observed. Normal (gray): individuals with no actin in the nucleus. Abnormal (black): individuals with actin in the nucleus. Source data are available for this figure: SourceData F4.

modification upon expression of Dam alone. The regions enriched with the resulting Dam-Lam signals were equivalent to LADs.

We averaged the data of two independent DamID experiments, which highly correlated with each other (Pearson correlation of 1 for the WT and 0.97 for *PIGB*[13]), when scored using 100 kb bins (Fig. S3 A). Chromatin domains enriched for Lamin Dm0 interactions were determined using the hidden Markov model (HMM) algorithm. Fig. S3, B and C (WT and *PIGB*[13], respectively) provides genome–NL interaction maps showing $\log_2$(Dam-Lam/Dam) profiles and HMM-determined domains of Lamin Dm0 enrichment. The total number and mean length of LADs were 89 and 61.6 kb in the WT, respectively, and 77 and 72.9 kb in *PIGB*[13], respectively (Fig. 5 A). Although the global pattern of interactions was similar in the WT and *PIGB*[13], visual inspection suggested that some interactions were weaker in the latter (Fig. 5 B). In particular, DamID scores were decreased in centromeres of chromosomes II and III (Fig. 5, B and C). This suggests that centrosome regions weakly interact with lamins in the WT and lose this affinity in *PIGB*[13]. To investigate whether there were any changes in centromere positioning, we attempted staining using anti-centromere antibodies. In cells like leg discs, one to two centromeres were detected in the nucleus. However, in muscle cells, signals were observed both in the cytoplasm and nucleus, making it difficult to determine whether centromeres in the nucleus were accurately labeled. Consequently, we could not ascertain whether there were any changes in centromere positioning (Fig. S3 D).

We next recalculated DamID scores without using bins to identify differences of small LADs between the WT and *PIGB*[13]. Correlations between samples were relatively low, with Pearson

correlations of 0.93 for the WT and 0.82 for *PIGB*[13] (Fig. S4 A). Genome–NL interaction maps and determined LADs for WT and *PIGB*[13] are shown in Fig. S4, B and C, respectively. The total number and mean length of LADs were 4,773 and 9.93 kb in the WT, respectively, and 7,666 and 6.29 kb in *PIGB*[13], respectively (Fig. 5 D), indicating there were more small LADs in *PIGB*[13]. An example is shown in Fig. S5 A. We classified LADs into three categories (Fig. 5 E), namely, cLADs, which were LADs common to the WT and *PIGB*[13]; fLADs, which were LADs in the WT that disappeared in *PIGB*[13]; and fiLADs, which were ectopic LADs that appeared in *PIGB*[13]. About 90% of LADs were cLADs in the WT, whereas about half of LADs were fiLADs in *PIGB*[13] (Fig. 5 F). This indicates that most LADs in the WT are conserved in *PIGB*[13], but *PIGB*[13] has almost as many additional LADs as cLADs, which is consistent with the increased number of LADs in *PIGB*[13] as mentioned above. fLADs were located in intergenic regions (128 sites) and intron and intron-containing regions (273 sites; Fig. S5 B), while fiLADs were located in eight times more intron and intron-containing regions (2,273 sites; Fig. 5 G). These results indicate that additional interactions between intron-containing chromatin regions and Lamin Dm0 occur in *PIGB*[13].

Gene ontology (GO) analysis using Metascape was performed to examine genes enriched in fiLADs. fLADs and fiLADs both contained genes involved in several developmental processes including muscle structure development, but fiLADs had larger fraction values and smaller logqscores (Fig. 5 H), indicating that they are enriched with these genes.

We examined whether gene expression was actually altered. As described later, a muscle defect was observed in *PIGB*[13]; therefore, we focused on 128 genes linked with muscle structure

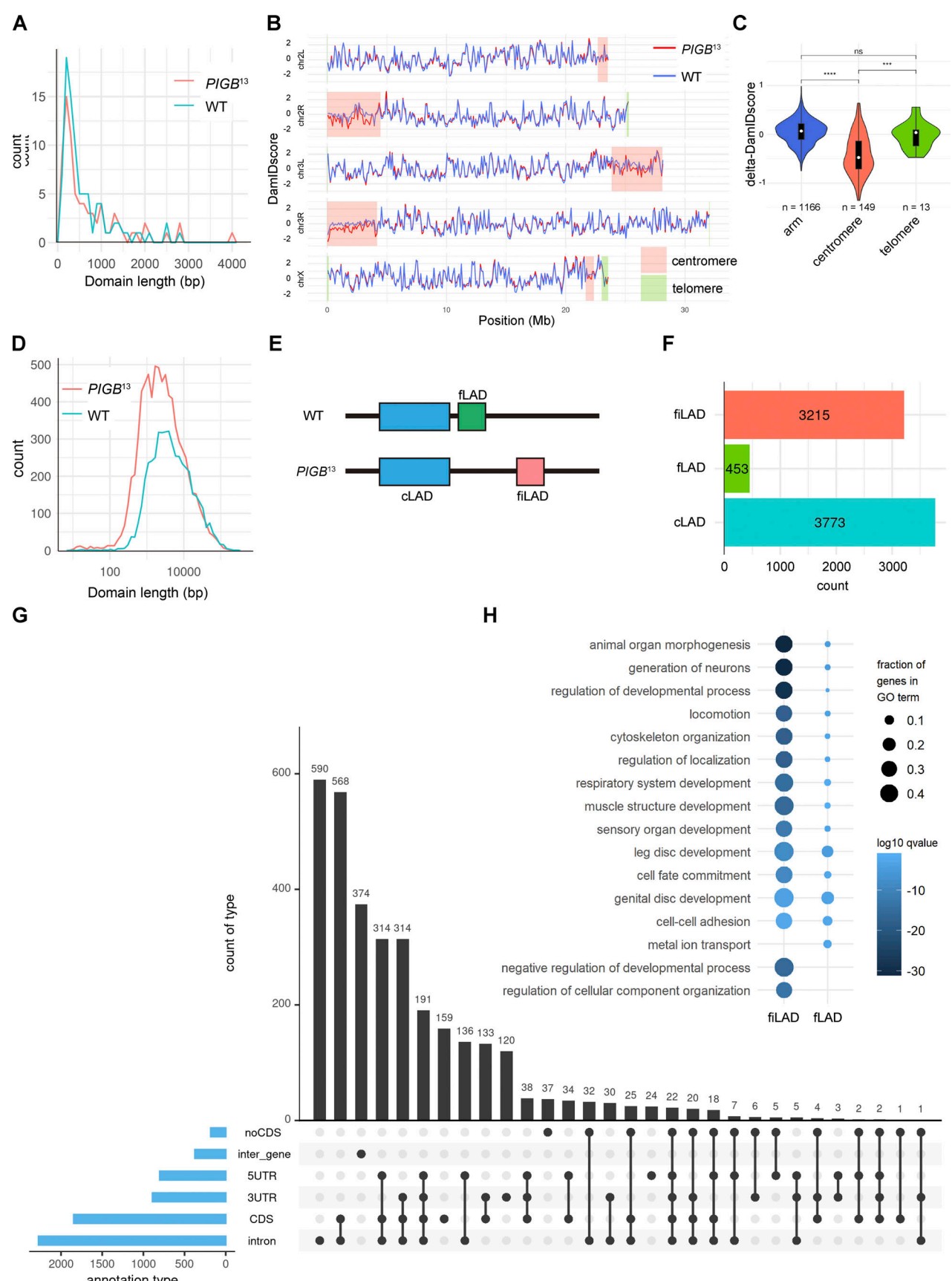

Figure 5. **Loss of PIGB affects formation of LADs. (A)** Distribution of LAD sizes in the WT (blue) and *PIGB*[13] (red) using 100 kb bins. **(B)** Line plots of DamID scores only in the WT (blue) and *PIGB*[13] (red). The DamID score was calculated by averaging the log₂ (Dam-Lam/Dam) across biological replicates for each

sample. Pink and green rectangles represent centromeres and telomeres, respectively. Peri-centromere and telomere regions were obtained by referring to the cytoband definition from the UCSC genome browser (Kent et al., 2002). **(C)** Violin plots comparing the frequency of contacts of arms, centromeres, and telomeres ($\log_2$[Dam-Lam/Dam]) in the WT versus $PIGB^{13}$. ***, P < 0.001; ****, P < 0.0001 by the Wilcoxon rank sum test. ns, not significant. **(D)** Distribution of LAD sizes in the WT (blue) and $PIGB^{13}$ (red) scored without using bins. **(E)** Diagram of cLADs, fLADs, and fiLADs as defined in this study. **(F)** Counts of cLADs, fLADs, and fiLADs. **(G)** Upset plot of fiLADs annotated as intron, CDS, 3UTR, 5UTR, inter_gene, and no CDS. **(H)** GO analysis of fiLADs and fLADs.

development (GO0061061) in fiLADs. Among them, we extracted genes whose defects were reported to cause phenotypes in muscle from FlyBase (http://flybase.org/), selected 12 genes with high expression levels using modENCODE Tissue Expression Data of FlyBase, and performed qPCR. Among the selected genes, specific primers could not be designed for the *myosin heavy chain* and *molecule interacting with CasL*. 6 of the 10 genes examined (*Zasp66*, *Prm*, *br*, *cher*, *how*, and *akirin*) were significantly downregulated in both types of $PIGB$-deficient larvae. Expression of *tn* was significantly decreased in $PIGB^{CRP5}$ but not in $PIGB^{13}$. Expression of *actn*, *cora*, and *ens* was unchanged (Fig. 6). These results suggest that changes in LADs affect the expression levels of some, but not all, genes.

### Loss of PIGB results in softened nuclei

The structural stability of nuclei is maintained by the structures of the cytoskeleton, lamins, and chromatin (Hobson and Stephens, 2020; Kalukula et al., 2022). The heterogenous distribution of Lamin Dm0 and change of LADs in $PIGB$-deficient larvae prompted us to examine the mechanical properties of nuclei using a dual microneedle-based setup that we previously developed (Shimamoto and Kapoor, 2012; Shimamoto et al., 2011, 2017). To visualize nuclei of body wall muscle, mCherry fused to the SV40 large T antigen nuclear localization signal (mCherry-NLS) was expressed in the WT and $PIGB^{13}$ using Mef2-Gal4. When third instar larvae were dissected and body wall muscle was exposed, nuclei were located on the cell surface and could therefore be captured by microneedles without penetrating the cell membrane (Fig. 7 A). The mechanical properties of nuclei in VO4 or VO5 muscle were thus measured in situ by pressing one microneedle located close to the other (black arrow, Fig. 7 A). The amount of applied force was monitored based on the deflection of the flexible, force-calibrated tip of one microneedle, and the resulting deformation that arose in the nucleus was

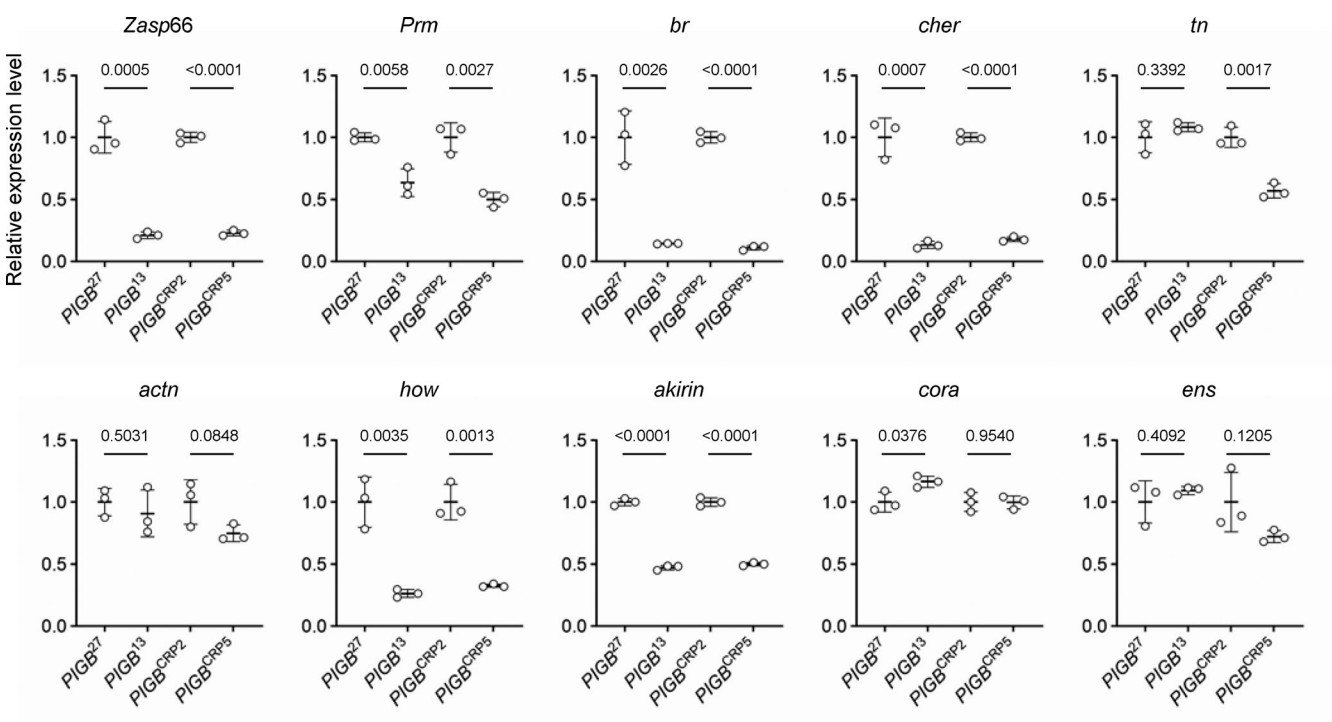

Figure 6. **Expression levels of genes in fiLADs in WT and *PIGB*-deficient larvae.** mRNA levels of genes linked with muscle structure development (GO0061061) in fiLADs with high expression levels in WT ($PIGB^{27}$ and $PIGB^{CRP2}$) and $PIGB$-deficient ($PIGB^{13}$ and $PIGB^{CRP5}$) larvae. The genes analyzed were *Z band alternatively spliced PDZ-motif protein 66* (*Zasp66*), *Paramyosin* (*Prm*), *broad* (*br*), *cheerio* (*cher*), *thin* (*tn*), *αactinin* (*actn*), *held out wings* (*how*), *akirin, coracle* (*cora*), and *ensconsin* (*ens*) as well as *ribosomal protein L32* (*rpl32*) as an internal control. Three batches of mRNA from 20 carcasses of each type of larvae were prepared (biological replicates = 3). These mRNA samples were subjected to the qPCR experiment three times (experimental replicates = 3). From the three experiments, the amounts of *rpl32* and each gene were calculated, and the amount of each gene normalized by that of *rpl32* was averaged. The average value was plotted for each biological replicate (white circle) normalized by WT as a control. The thick black horizontal bar and thin gray horizontal bar show the mean and SD (biological replicates = 3), respectively. The number at the top of the graph is the P value (biological replicates = 3) calculated using the unpaired two-tailed *t* test. The actual values, means, and SD are shown in Table S1.

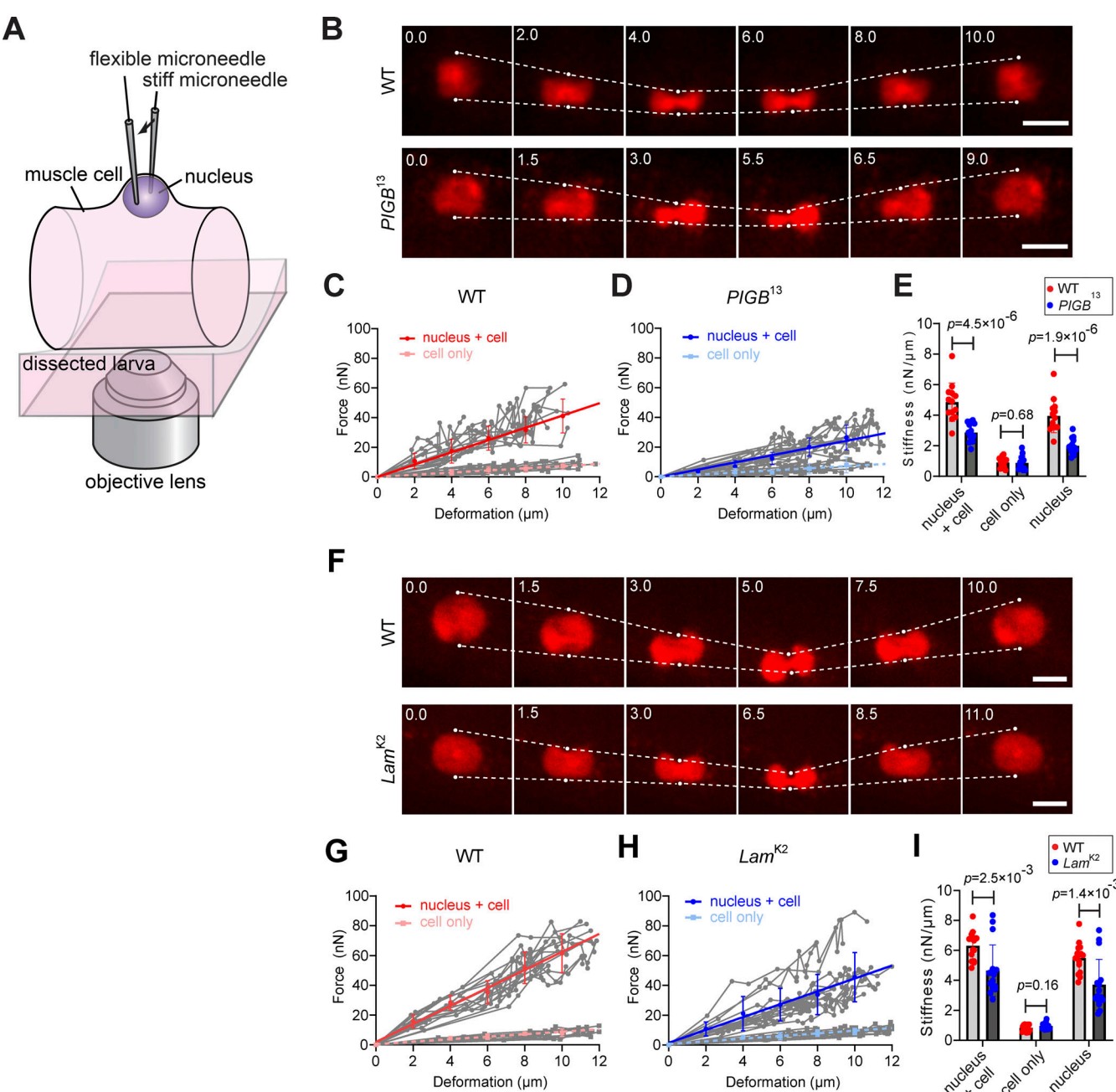

Figure 7. **PIGB is required to maintain nuclear stiffness. (A)** Schematic showing the microneedle-based setup used to analyze nuclear stiffness. A single nucleus of a fly larva muscle cell (VO4 or VO5) is captured and compressed using a pair of glass microneedles. One microneedle is rigid and used to apply controlled deformation to the nucleus (black arrow). The other microneedle is flexible and stiffness-calibrated such that the force that acts on the nucleus can be measured. **(B–I)** Deformability of nuclei in WT and mutant larvae. Measurements were performed with Mef2>mcherryNLS, $PIGB^{13}$ ($PIGB^{13}$) (B–E) or Mef2>mcherryNLS, $Lam^{K2}$ ($Lam^{K2}$) (F–I), along with the Mef2>mcherryNLS (WT) strain prepared on the same day in each experiment. **(B and F)** Representative time-lapse images. Nuclei (red, mCherry) were pressed by moving the stiff microneedle (upper white dots) while the flexible, force-calibrated microneedle was displaced from the equilibrium point (lower white dots). Timestamps are in seconds. Scale bars, 10 µm. **(C, D, G, and H)** The force-deformation plots in the WT ($n = 13$ in C and 15 in G), $PIGB^{13}$ ($n = 14$) (D), and $Lam^{K2}$ ($n = 16$) (H). Measurements were first performed by pressing the nucleus via cell membrane structures (circles; labeled "nucleus + cell") and then at locations lacking nuclei to estimate the non-nuclear contribution (squares; labeled "cell only"). Gray plots are data from individual nuclei. Colored plots are mean ± SD at each 2 µm bin. Slopes were determined by linear regression ($R^2 > 0.96$). **(E and I)** Comparison of nuclear stiffness between the WT and either $PIGB^{13}$ (E) or $Lam^{K2}$ (I). Red, WT ($n = 13$ and 15 in E and I, respectively); blue, $PIGB^{13}$ ($n = 14$) (E) or $Lam^{K2}$ ($n = 16$) (I). The two right-most columns are the values after subtracting the non-nuclear contribution. Bars are SD. P values were determined by the Mann–Whitney $U$-test.

observed in mCherry channel images. Fig. 7 B shows typical time-lapse imaging data for the WT and $PIGB^{13}$. Nuclei exhibited micron-sized deformation upon pressing, and the deformation was restored as soon as the applied force was released, showing

a predominantly elastic response. Neither WT nor $PIGB^{13}$ nuclei exhibited blebbing or rupture. On the other hand, the deformability of nuclei was significantly greater in $PIGB^{13}$ than in the WT. This was quantitatively revealed in the force-deformation

plot, which was obtained by repeating several press-release cycles with different force magnitudes. The linear regression slope, which indicates the structure's stiffness, was approximately twofold smaller in $PIGB^{13}$ than in the WT (2.4 versus 4.2 nN/μm; $n$ = 14 and 13, respectively; solid lines, Fig. 7, C and D). To estimate the influence of non-nuclear structures surrounding the nucleus (e.g., cell membrane structures), the same micromanipulation procedure was performed at a region close to but lacking nuclei. The magnitude of the resistance force was more than three times smaller in the absence of nuclei than in the presence of nuclei, and the calculated slopes did not significantly differ between the WT and $PIGB^{13}$ (0.73 and 0.71 nN/μm, respectively; dashed lines, Fig. 7, C and D). Consistently, the stiffness of individual nuclei, as determined by linear regression of the force-deformation plot obtained from each sample, was significantly higher in the WT than in $PIGB^{13}$ (4.9 ± 1.3 versus 2.9 ± 0.6 nN/μm, P = 4.5 × $10^{-6}$ by the Mann–Whitney $U$-test; "nucleus + cell," Fig. 7 E). The result was consistent when the non-nuclear contribution was taken into account (4.0 ± 1.1 versus 2.0 ± 0.5 nN/μm, P = 1.9 × $10^{-6}$ by the Mann–Whitney $U$-test; "nucleus," Fig. 7 E). This clearly showed that $PIGB$ deficiency resulted in softened nuclei.

Next, we investigated whether Lamin Dm0 affects nuclear stiffness. Time-lapse imaging revealed the predominant elasticity of nuclei of the *Lamin Dm0* mutant (*Lam*$^{K2}$), similar to the WT and $PIGB^{13}$ (Fig. 7 F). The slope of the force-deformation plot indicated that nuclear stiffness was lower in *Lam*$^{K2}$ than in the WT (4.3 vs. 6.1 nN/μm, solid lines, Fig. 7, G and H; $n$ = 15 and 16, each from four larvae, respectively). The contribution of non-nuclear materials was comparably small as in $PIGB^{13}$ (0.84 and 0.89 nN/μm for the WT and *Lam*$^{K2}$, respectively; dashed lines, Fig. 7, G and H). The stiffness of individual nuclei was 6.3 ± 0.9 and 4.7 ± 1.7 nN/μm in the WT and *Lam*$^{K2}$, respectively (5.5 ± 1.0 and 3.7 ± 1.7 nN/μm, respectively, after subtracting the non-nuclear contribution), indicating that nuclei of the *Lamin Dm0* mutant were more deformable than those of the WT (P = 2.5 × $10^{-3}$ by the Mann–Whitney $U$-test; Fig. 7 I). For these measurements, the WT and mutant samples (either $PIGB^{13}$ or *Lam*$^{K2}$) were prepared in parallel using an identical procedure and alternatively subjected to measurement within the same time window so that day-to-day variation in measured nuclear stiffness was minimized. Together, these results suggest that nuclear stiffness is determined not only by the existence of Lamin Dm0 but also by its uniform distribution, which is dependent on PIGB.

### PIGB mutation impairs muscle structure independently of GPI synthesis
Finally, we addressed the biological significance of the NE localization of PIGB in the muscle. We observed the muscle structure using phalloidin. WT and $PIGB$-deficient larvae within 12 h of the wandering stage ($PIGB^{27}$ [<12 h], $PIGB^{CRP2}$ [<12 h], $PIGB^{13}$ [<12 h], $PIGB^{CRP5}$ [<12 h], and $PIGB^{13}$ [<12 h]) exhibited typical striated muscle patterns (Fig. 8 A, upper and middle panel, and Fig. S6). However, PIGB-deficient larvae 2 d later ($PIGB^{13}$ [>2 d] and $PIGB^{CRP5}$ [>2 d]) displayed cracks in the cell surface (Fig. 8 A, lower panel, and Fig. S6). Fig. S6 provides

representative images of muscle in WT and $PIGB$-deficient larvae. This defect was frequently observed in VL1 and VL2 muscles. The proportion of larvae exhibiting such defects was 0% and 8.3% in $PIGB^{27}$ [<12 h] and $PIGB^{CRP2}$ [<12 h], both of which are WT, respectively ($n$ = 12). By contrast, the proportion of larvae exhibiting such defects was 8.3% in both $PIGB^{13}$ [<12 h] and $PIGB^{CRP5}$ [<12 h], but was 66.7% and 58.3% in $PIGB^{13}$ [>2 d] and $PIGB^{CRP5}$ [>2 d], respectively ($n$ = 12; Fig. 8 B). We further investigated whether these phenotypes depend on GPI synthesis. Cracked muscles were observed in 33.3% ($n$ = 36) of 2-d-old PIGB-deficient larvae ($PIGB^{13}$ [>2 d]) with Mef2-Gal4 under a condition where they were observed in only 2.7% of Mef2-Gal4 [<12 h]. When wtPIGBmyc or ΔactPIGBmyc was expressed in $PIGB^{13}$ using Mef2-Gal4, 0% ($n$ = 36), 10.1% ($n$ = 36), and 5.6% ($n$ = 36) of $PIGB^{13}$ larvae expressing wtPIGBmyc (68A4) [<12 h], ΔactPIGBmyc (68A4) [>2 d], and ΔactPIGBmyc (55C4) [>2 d] exhibited such a defect, respectively (Fig. 8 C). Cracked muscles were also rescued by ERPIGB (68A4). However, we could only observe early larvae [<12 h], and therefore it remains uncertain whether the rescue was fully achieved. These data indicate that the function of PIGB at the NE affects muscle architecture independently of GPI synthesis.

## Discussion
PIGB was originally identified as an enzyme that catalyzes GPI synthesis. In this study, we showed that PIGB is a multifunctional protein involved in the formation and maintenance of the NL in *Drosophila*, including NL-localized protein distribution, LAD organization, and nuclear membrane stiffness (Fig. 9). We also showed that the proper distribution of NL-localized proteins requires localization of PIGB to the NE, rather than its GPI synthesis activity (Fig. 4, D and E). To the best of our knowledge, the absence of NE-localized proteins such as SUN proteins and LEM domain proteins did not disrupt the lamin distribution, underscoring that PIGB is a unique protein responsible for maintaining the uniform lamin meshwork (Fig. 2 F). Furthermore, *PIGB*-deficient skeletal muscle is disrupted, suggesting that a well-organized NL is essential for proper cell structure and function. This study sheds light on the underlying mechanisms and biological significance of the formation of ordered nuclear lamin meshworks.

### Abnormal distribution of NL proteins in the PIGB mutant
In the NE of the larval wall muscle of $PIGB^{13}$, Lamin Dm0, the LEM domain protein Ote, and NPCs had a heterogeneous distribution, indicating that the NL is disorganized in this mutant. It has recently been reported that oxidoreductases and lipid synthases, which play a role in regulating cell growth and survival, are also localized in the NL (Cheng et al., 2023), and their localization and function may be abnormal in *PIGB* mutants. We previously reported that PIGB binds to Lamin Dm0 (Yamamoto-Hino et al., 2020), and overexpression of PIGB in this study affected the localization of Lamin Dm0. These results suggest that disruption of the NL in $PIGB^{13}$ is due to the impaired distribution of Lamin Dm0. Furthermore, as far as we have found, PIGB is the only nuclear membrane protein whose deficiency leads to the

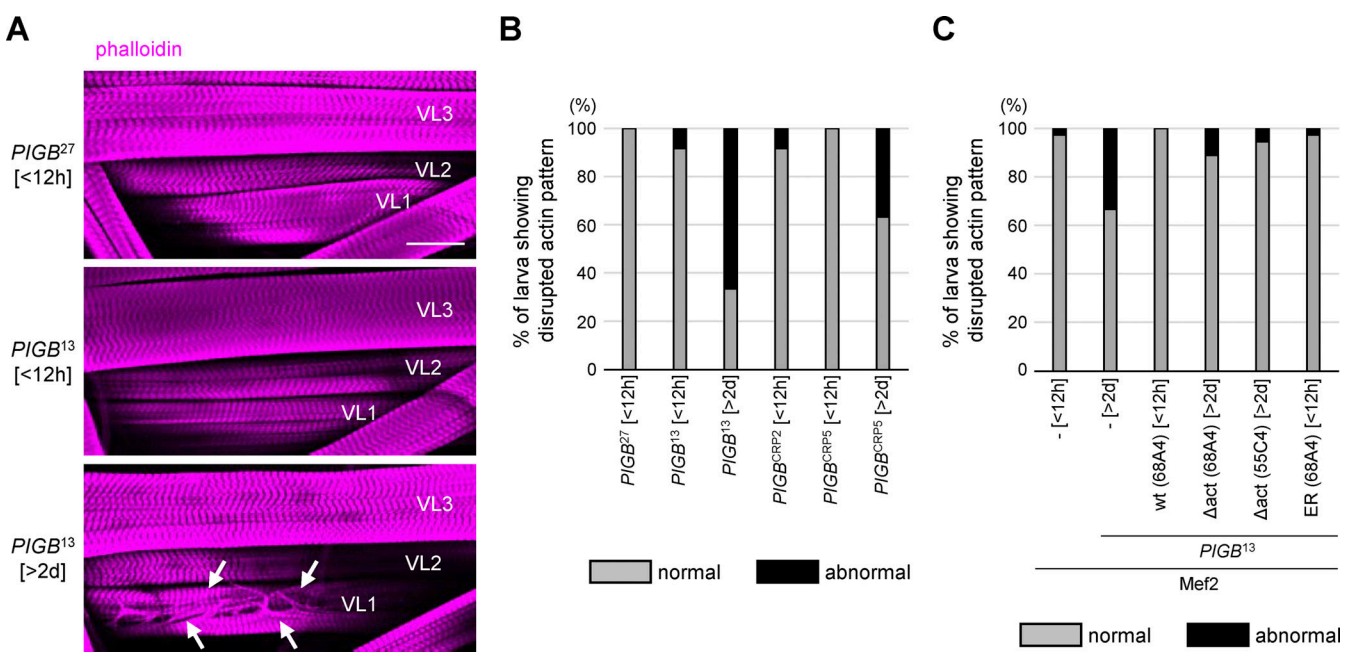

**Figure 8.  PIGB affects muscle structure independently of GPI synthesis. (A)** Phalloidin staining of larval wall muscle in *PIGB*²⁷ [<12 h] (upper), *PIGB*¹³ [<12 h] (middle), and *PIGB*¹³ [>2 d] (lower). Arrows indicate cracks in VL1 in *PIGB*¹³ [>2 d]. Bar, 500 µm. **(B)** Percentage of larvae showing cracks in skeletal wall muscle in *PIGB*²⁷ [<12 h], *PIGB*¹³ [<12 h], *PIGB*¹³ [>2 d], *PIGB*ᶜᴿᴾ² [<12 h], *PIGB*ᶜᴿᴾ⁵ [<12 h], and *PIGB*ᶜᴿᴾ⁵ [>2 d]. 12 individuals per strain were observed. Normal (gray): individuals with typical striated muscle. Abnormal (black): individuals with a disrupted actin pattern at the muscle cell surface. **(C)** Percentage of larvae showing cracks in wall skeletal muscle in Mef2-Gal4 only (Mef2, –, [<12 h]), Mef2-Gal4, *PIGB*¹³ (Mef2, *PIGB*¹³ -, [>2 d]), and *PIGB*¹³ expressing wtPIGBmyc (Mef2, *PIGB*¹³, wt (68A4) [<12 h]), 3UAS-ΔactPIGBmyc (68A4) ([Mef2, *PIGB*¹³, Δact [68A4] [>2 d]), 20UAS-ΔactPIGBmyc(55C4) ([Mef2, *PIGB*¹³, Δact [55C4] [>2 d]), and 3UAS-ERPIGBmyc (68A4) ([Mef2, *PIGB*¹³, ER [68A4] [<12 h]) using Mef2-Gal4. 36 individuals per strain were observed. Normal (gray): individuals with typical striated muscle. Abnormal (black): individuals with a disrupted actin pattern at the muscle cell surface.

abnormal distribution of Lamin Dm0. Recent cryo-electron tomography observations suggested that nuclear membrane proteins and chromatin are located in the spaces between the lamin meshes (Turgay et al., 2017) and that these meshes might be simply spaces pushed apart by these proteins and chromatin.

However, our results revealed that PIGB is required to keep the lamin meshwork uniform.

How does PIGB keep the lamin mesh structure uniform? It is possible that PIGB affects the assembly of Lamin Dm0. At least in vitro, bacterially expressed mammalian and *Drosophila* lamins

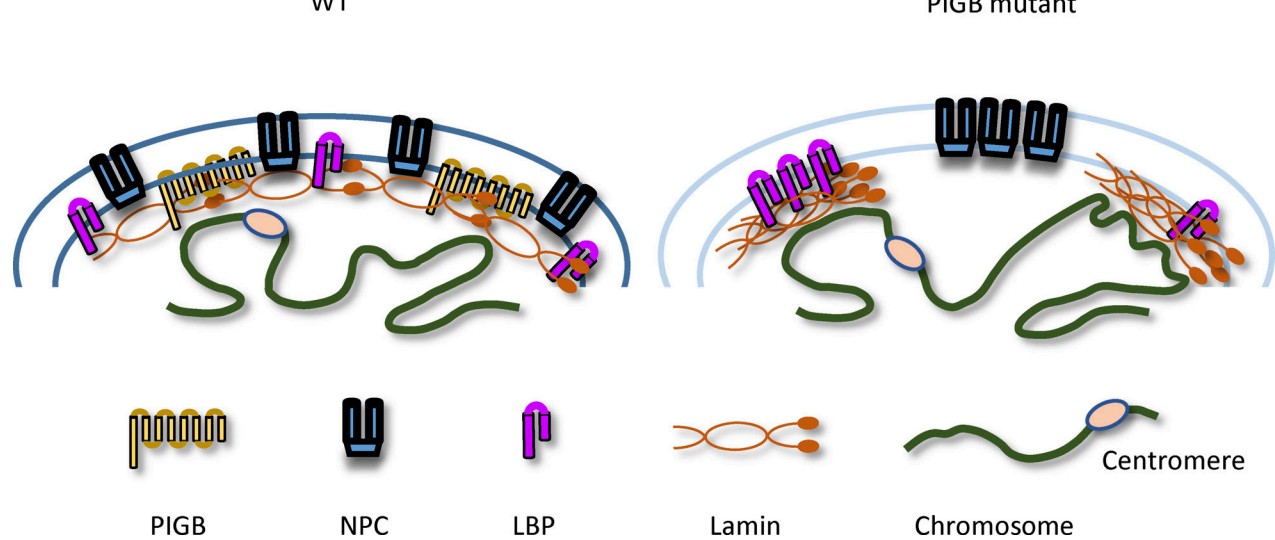

**Figure 9.  Schema of the NL in *PIGB*-deficient nuclei.** WT nuclei. Lamin Dm0, NPCs, and lamin-binding proteins are homogenously distributed and well-organized at the nuclear periphery. By contrast, in *PIGB*-deficient nuclei, Lamin Dm0 and lamin-binding proteins are irregularly accumulated, and NPCs are clustered. In addition, small LADs are increased, and centromeres may migrate away from the nuclear periphery.

tend to aggregate into inclusion bodies that need to be dissolved with 8 M urea (de Leeuw et al., 2018). In addition, refolded lamins assemble into large paracrystalline arrays rather than into fibers, which does not occur in vivo (Stuurman et al., 1998). This suggests that homogeneous lamin filament assembly requires specific chemical conditions and/or some factors located in the NE. It was reported that an artificially designed ankyrin repeat protein prevents lamin aggregation upon coexpression with Lamin A in bacteria (de Leeuw et al., 2018; Zwerger et al., 2015). Although it is unclear whether Lamin Dm0 assembles correctly in *PIGB*[13], it is conceivable that PIGB is one of the factors required for higher-order assembly of lamins. Moreover, the C-terminal Ig-fold regions of lamins are observed at 20 nm intervals on both sides of filaments (Turgay et al., 2017). We have reported that the Ig-fold domain of Lamin Dm0 mainly binds to PIGB, while the other domains may slightly contribute to the association with PIGB (Yamamoto-Hino et al., 2020). It is possible that PIGB acts as a support for Lamin Dm0 to spread the network homogenously with proper spacing via its C-terminal Ig-fold region. It is noteworthy that the C-terminal Ig-fold region of lamins contains the binding sites for most lamin-binding proteins including NPCs (Dechat et al., 2008). PIGB may maintain the uniform lamin meshwork by regulating the associations between Lamin Dm0 and lamin-binding proteins.

Interestingly, a recent report revealed that the spatial distribution of NPCs is lamin-dependent, and conversely, knockdown of several lamin-associated nucleoporins results in clustering of NPCs and expansion of the Lamin A meshwork (Kittisopikul et al., 2021). In *PIGB*-deficient mutants, the localizations of Lamin Dm0 and NPCs appear to be complementary, indicating that NPCs do not associate with Lamin Dm0. Loss of Lamin A/C or Lamin B1 causes clustering of NPCs, suggesting that NPCs tend to cluster by themselves. It is possible that NPCs not associated with Lamin Dm0 in the *PIGB* mutant form clusters and expand the network space of Lamin Dm0.

In addition to farnesylation, other posttranslational modifications of lamins also affect their localizations (Murray-Nerger and Cristea, 2021). Human Lamin A can be SUMOylated at lysine 201. The disease-inducing Lamin A mutants E203G, E203K, and K201R exhibit decreased SUMOylation and have a punctate, aberrant subcellular localization at the nuclear periphery (Zhang and Sarge, 2008). Lamin Dm0 can also be SUMOylated by *Drosophila* UbL and SUMO E2 conjugase homolog, Smt3 and Lesswright, respectively, when overexpressed in cultured cells (Pirone et al., 2017). However, the amino acid of Lamin Dm0 that corresponds to lysine 201 of human Lamin A is arginine, which is not a substrate for SUMOylation, suggesting that other amino acids are SUMOylated. Our immunoblotting with an anti-Lamin Dm0 antibody found no difference between *PIGB*[27] and *PIGB*[13] even upon overexposure (Fig. S1 J); however, it would be interesting to investigate if PIGB affects posttranslational modifications of Lamin Dm0 and if this is involved in the formation of a uniform meshwork of Lamin Dm0.

### Influence of PIGB on LAD formation
LADs were altered in *PIGB*-deficient muscle. It is unclear whether this is due to a direct effect of PIGB on chromatin or the abnormal distribution of Lamin Dm0. Although it has been reported that the absence of all lamins in mouse embryonic stem cells does not affect the overall structure of LADs (Amendola and van Steensel, 2015), lamin proteins are assumed to tether LADs to the NL (Manzo et al., 2022). Knockdown of Lamin Dm0 in *Drosophila* S2 cells results in the compaction of chromatin and its repositioning away from the NE (Ulianov et al., 2019). Knockout of Lamin B1 in human breast cancer cells leads to detachment of LADs from the nuclear periphery, accompanied by global redistribution and decompaction of chromatin (Chang et al., 2022). These data suggest that B-type lamins affect the establishment and maintenance of LADs. The altered distribution of Lamin Dm0 in *PIGB*[13] may affect the architecture of LADs.

The interactions between centromeres of chromosomes II and III and Lamin Dm0 were reduced in *PIGB*[13]. In *Drosophila* cultured S2 cells and hemocytes, centromeres are positioned and clustered near the nucleolus (Padeken et al., 2013). However, Pindyurin et al. reported that LADs are significantly more variable in different tissues of *Drosophila* than in mammals, with centromeres located closer to the NL in neurons than in Kc167 cultured cells of embryonic origin (Pindyurin et al., 2018). They also found that heterochromatin protein 1a (HP1a) does not overlap with LADs in Kc167 cells, whereas in terminally differentiated cells such as neurons, glia, and adipocytes, the HP1a domain strongly overlaps with LADs and appears to repress the expression of genes that are not required for differentiation. We attempted to determine if centromeres are positioned at the nuclear periphery in body wall skeletal muscle and if this is altered in *PIGB*[13] using an anti-CID antibody but could not detect centromeres in the muscle (Fig. S3 D). DamID analysis revealed that centromeres were located closer to the NE in the skeletal muscle of the WT than of *PIGB*[13], suggesting that movement of centromeres away from the NE in *PIGB*[13] affects gene expression.

Our DamID-seq analysis without using bins revealed that lamin-associated small introns (fiLADs) were increased in *PIGB*[13]. GO analysis showed that genes involved in developmental processes were enriched in fiLADs. LADs vary according to the cell type, tissue type, and developmental stage, and repress the expression of unnecessary genes (Pindyurin et al., 2018; Shah et al., 2021). In fact, the expression of some genes involved in muscle structure development in fiLADs was downregulated (Fig. 6). Larval wall skeletal muscle of *PIGB*[13] appeared to develop normally but was deformed at the third instar larval stage. This probably reflects changes in gene expression in fiLADs. To confirm this conclusion, comprehensive analysis is needed to determine whether there are changes in gene expression that correspond to changes of LADs. However, it is difficult to extract changes in gene expression in *PIGB* mutants caused by alteration of LADs because these changes reflect complex factors, namely, the absence of GPI-anchored proteins and cell non-autonomous effects.

### PIGB contributes to nuclear stiffness
The present study showed that stiffness of the nuclear membrane was reduced in *PIGB*[13]. PIGB may support the nucleus as a structural protein of the nuclear membrane, but it is widely proposed that the physical properties of the nuclear membrane

are determined by the NL and chromatin structures (Hobson and Stephens, 2020; Kalukula et al., 2022). Regarding the role of lamins in the physical properties of nuclear membranes, the stoichiometry of Lamin A:Lamin B correlates with nuclear stiffness in various tissues and at various differentiation stages (Pajerowski et al., 2007; Swift et al., 2013). Early experiments showed that depletion of Lamin A/C softens nuclei and makes them more deformable. Furthermore, it has been reported that the expression of mutant Lamin C, which induces laminopathy, in *Drosophila* muscles can lead to mechanical stress-induced deformations in nuclear shape (Shaw et al., 2022; Zwerger et al., 2013). These reports suggest that Lamins A and C are important contributors to the mechanical stiffness of nuclei (Broers et al., 2004; Lammerding et al., 2004, 2006). However, it has been reported that depletion of Lamin B1 in HEK293 cells, which express low levels of Lamin A/C, increases nuclear stiffness (Stephens et al., 2017). In addition, a recent study showed that Lamin B1 also contributes to nuclear stiffness, while viscosity is mainly dictated by Lamin A (Wintner et al., 2020). These results suggest that Lamin B also affects the physical properties of the nuclear membrane. We found that *Lamin Dm0*-deficient nuclei were more deformable than WT nuclei, which indicates that the B-type Lamin contributes to nuclear stiffness in *Drosophila*. Moreover, *PIGB*-deficient nuclei with a disrupted Lamin Dm0 meshwork were softer than WT nuclei. This suggests that in addition to lamin levels, the uniform distribution of Lamin Dm0 is important to maintain nuclear rigidity. Changes in the local physical properties of the nuclear membrane are also of interest. For example, do the physical properties of the nuclear membrane differ between areas with and without accumulated lamins? In addition to lamins, chromatin acts as a viscoelastic component (Pajerowski et al., 2007) and is directly related to nuclear deformability (Furusawa et al., 2015; Mazumder et al., 2008). We have reported that nuclei with condensed chromatin possess significant elastic rigidity, whereas nuclei with decondensed chromatin are considerably soft (Shimamoto et al., 2017). Furthermore, nucleosome–nucleosome interactions via histone tails in chromatin act as a "nuclear spring" to suppress nuclear deformation (Maeshima et al., 2018). As mentioned earlier, the NL and chromatin are bound at LADs. It has been shown experimentally that this association is relevant for nuclear stability (Schreiner et al., 2015). In nuclei of *PIGB* mutants, LADs are also altered, which may affect nuclear stiffness together with the chromatin status.

Lamin Dm0 was heterogeneously distributed in muscle, but not in wing discs, nerves, and fat bodies of *PIGB*[13] (Fig. S1 I). The role of PIGB in nuclear function in tissues other than muscle is unknown. Muscles are subject to repeated stretching and contraction, and these mechanical stimuli may cause the irregular distribution of Lamin Dm0. It has been reported that *Lamin Dm0* null mutant larvae exhibit reduced locomotion, the absence or slight reduction in the size of LO1 muscle, and fine structure defects in fibrillar organization (Muñoz-Alarcón et al., 2007). Moreover, expression of *Drosophila* N-terminal truncated Lamin C or mutations representing human Lamin A laminopathy mutations in larval wall muscle result in lethality, mislocalization of nuclear membrane proteins, nucleo-cytoskeletal uncoupling,

and locomotion defects (Coombs et al., 2021; Dialynas et al., 2010, 2012; Schulze et al., 2009; Shaw et al., 2022; Zwerger et al., 2013). The fact that the phenotype is observed only in muscle upon *PIGB* deficiency may be related to the fact that even in laminopathies caused by lamin mutations, the tissues in which the pathology manifests itself differ depending on the type of mutation. Recently, it has been reported that mammalian PIGB is not only localized to the ER but also to the NE (Cheng et al., 2023). Investigating the molecules involved, including PIGB, in the establishment of a consistent lamin network would greatly contribute to nuclear biology and increase understanding of the pathogenesis of laminopathies.

## Materials and methods

### Fly stocks
*Drosophila* stocks (Table S2) were maintained using standard protocols and were fed a standard diet consisting of glucose (045-31167; FUJIFILM Wako Chemicals), cornmeal (Oriental Yeast), yeast (28957; Mitsubishi Corporation Life Sciences), and agar (260-01705; Kishida Chemicals). Experimental crosses were kept at 25°C.

### Generation of PIGB and *LBR* null mutant flies using CRISPR/Cas9-mediated gene editing
To generate the other PIGB null mutant allele, PIGB-specific oligonucleotides (5′-GGTCCAAACATATTACGTTC-3′ and 5′-GAACGTAATATGTTTGGACC-3′ [60 bp downstream of the first ATG]) were annealed and introduced into pBFv-U6.2 (National Institute of Genetics) to generate the U6-PIGB gRNA vector. The U6-PIGB-gRNA vector was injected into y1 w67c23; P{CaryP} attP1 embryos and integrated into the 55C4 genome site using the phiC31 integrase system at BestGene Inc. Offspring (in which the transgene was balanced) were collected to establish a stock. The PIGB-gRNA strain was then crossed with the *y2 cho2 v1*; *attP40[nos-Cas9]* strain, which expresses the Cas protein in germ cells. Resulting founder flies were crossed with balancer flies, and the sequence of each offspring was checked by DNA sequencing.

To generate the *LBR* null mutant allele, we chose two gRNAs, namely, 5′-GTTGACCCTTCTATTTTCTG-3′ (8 bp downstream of the stop codon) and 5′-GCGGAATGTACAGTTGCGGC-3′ (144 bp upstream of the first ATG of LBR-RC) using the flyCRISPR target finder tool (http://targetfinder.flycrispr.neuro.brown.edu/). The oligonucleotides were annealed and introduced downstream of the U6.1 and U6.3 promoters in pCFD4 (Port et al., 2014; 49411; Addgene), respectively. To construct a donor vector, 500–600 bp regions upstream of the first ATG and downstream of the stop codon and the lox-ey-DsRed-lox sequence from pPVxRF3 (138385; Addgene, provided by the National Institute of Genetics) were introduced into pBluescript. The primers used in this study are listed in Table S3. The mixture of gRNA vector and donor vector was injected into 200–300 embryos of the *y,w;;nos-Cas9(III-attP2)* strain at BestGene Inc. DsRed-positive G1 flies were backcrossed with *y,w* flies. DsRed-positive G2 flies were backcrossed with *y,w* flies again, and then the edited gene was balanced to establish the stock line.

## Generation of the ΔactPIGBmyc construct and ΔactPIGBmyc-expressing flies

The ΔactPIGBmyc expression vector was produced by Quik-Change site-directed mutagenesis (200519; Agilent) to replace GATGAA (DE) at position 84–89 with CTGCCT (AA) in the PIGBmyc-pRmHa S2 cell expression vector. The ΔactPIGBmyc fragment was amplified and introduced into pcDNA3 (Invitrogen) for mammalian cell expression and into pJFRC4-3UAS or pJFRC7-20UAS for transgenic fly generation (26217 and 26220, respectively; Addgene). The expression vector pJFRC4-3UAS-ΔactPIGBmyc was injected into y1 w67c23; P[CaryP]attP2 embryos and integrated into the 68A4 genome site using the phiC31 integrase system at BestGene Inc. The expression vector pJFRC7-20UAS-ΔactPIGBmyc was injected into y1 w67c23; P[CaryP]attP1 embryos and integrated into the 55C4 genome site using the phiC31 integrase system at BestGene Inc. Offspring (in which the transgene was balanced) were collected to establish a stock.

wtPIGBmyc and PIGBmyc variants were expressed using the Gal4-UAS system. This system involves crossing the Mef2-Gal4 driver, which expresses the Gal4 transcription factor in a muscle-specific pattern, to the responder possessing a transgene driven by a UAS element. The UAS element is bound by Gal4, which results in the activation of transgene expression. The effects of expression of the responder gene in a muscle-specific pattern were assayed in the resulting progeny.

## Cell culture and plasmid transfection

S2 cells (purchased from Drosophila Genomics Resource Center) were cultured in Schneider's medium (21720024; Thermo Fisher Scientific) supplemented with 10% fetal calf serum (S-FBS-NL-015; SERANA) and penicillin–streptomycin (15140122; Thermo Fisher Scientific). In total, $5 × 10^5$ cells were seeded into a 24-well culture plate directly for immunoblotting or with a poly-L lysine (P4707; Sigma-Aldrich)-coated coverglass for immunostaining and transfected with 3 μg of plasmid using calcium phosphate transfection reagent (K278001; Thermo Fisher Scientific) after 6 h. After 24 h, the medium was changed to a normal culture medium. The next day, protein expression was induced by incubating cells in the presence of 0.1 mM $CuSO_4$ for 8 h. CHO-K1 (F21.3.8) and PIGB-deficient (CHO1.33) cell lines (Yamamoto-Hino et al., 2018) were cultured in Ham's F-12 medium (087-08335; FUJIFILM Wako Chemicals) supplemented with 10% fetal calf serum (S-FBS-NL-015; SERANA) and penicillin–streptomycin (15140122; Thermo Fisher Scientific). For immunoblotting, $1 × 10^5$ PIGB-deficient (CHO1.33) cells were seeded into a 24-well plate. The next day, 0.8 μg of wtPIGBmyc/pcDNA3, ΔactPIGBmyc/pcDNA3, and pcDNA3 were transfected using 2 μl of Lipofectamine 2000 (11668027; Thermo Fisher Scientific). After 24 h, cells were harvested. For fluorescence-activated cell sorting (FACS) analysis, 0.64 μg of wtPIGBmyc/pcDNA3, ΔactPIGBmyc/pcDNA3, or pcDNA3 was simultaneously transfected with 0.16 μg of an EGFP expression plasmid (EGFP/pcDNA3) using 2 μl of Lipofectamine 2000 (11668027; Thermo Fisher Scientific). After 2 d, cells were harvested and subjected to FACS analysis.

## Antibodies

The antibodies used in this study are listed in Table S4.

## Immunostaining

For the standard preparation of larval body wall muscle, a longitudinal cut was performed along the ventral midline of the larva. Organs were subsequently removed, leaving behind a tissue termed as a carcass, consisting of body wall muscles attached to tendon cells and adhered to the hypodermis. This carcass was then subjected to the following experiments. Third instar larvae were dissected in phosphate-buffered saline (PBS) and fixed in 4% paraformaldehyde (161-20141; FUJIFILM Wako Chemicals) for 30 min at room temperature. After three washes with PBS containing 0.2% Triton X-100, specimens were blocked in PBS containing 0.5% bovine serum albumin (BSA; 01863-48; Nacalai Tesque) and 0.2% Triton X-100 (T8787; Sigma-Aldrich) and incubated with the appropriate primary antibodies overnight at 4°C. After three washes with PBS containing 0.2% Triton X-100, specimens were stained with secondary antibodies, DAPI (62248; Thermo Fisher Scientific), and fluorescently labeled phalloidin for 3 h at room temperature. After three washes with PBS containing 0.2% Triton X-100, specimens were mounted using Vectashield (H-1000; Vector Laboratories). Cultured cells were fixed in 4% paraformaldehyde for 30 min at room temperature. After three washes with PBS containing 0.05% Triton X-100, cells were incubated with the appropriate primary antibodies in PBS containing 0.5% BSA for 2 h at room temperature. After three washes with PBS containing 0.05% Triton X-100, cells were stained with secondary antibodies and DAPI for 1 h at room temperature. After three washes with PBS containing 0.05% Triton X-100, cells were mounted using Vectashield.

## Capture of fluorescence images

All images were captured under a laser-scanning confocal microscope (LSM710 and LSM980; Zeiss) operated with ZEN software (Zeiss). Dyes and fluorescent proteins were sequentially excited at their respective excitation maxima with a diode laser and fluorescence was detected by a filter adjusted to the respective dye emission maxima as follows: DAPI: 405 nm, 410–585 nm; Alexa Fluor 488: 488 nm, 493–543 nm; Cy3/phalloidin: 561 nm, 566–624 nm; and Alexa Fluor 633: 639 nm, 645–757 nm.

In Fig. 1 A and Fig. 3 A, images were captured with a PlanAPOCHOMAT 10×/0.45 M27 objective. Data were acquired with a view field of 512 × 512 pixels, a pixel size of 1.28 μm × 1.28 μm, and a pinhole size of 20 μm.

In Fig. 8 A and Fig. S6, images were captured with a PlanAPOCHOMAT 20×/0.8 M27 objective. Data were acquired with a view field of 1,024 × 1,024 pixels, a pixel size of 0.414 μm × 0.414 μm, and a pinhole size of 3 μm.

High magnification images (Fig. 1, B and D; Fig. 2, A, C, and F; Fig. 3, C and D; Fig. 4, C and D; Fig. S1, C, E, G, I, and L; Fig. S3 D) were captured with a PlanAPOCHOMAT 40×/1.4 Oil DIC M27 objective, a view field of 1,024 × 1,024 pixels, a pixel size of 0.1 μm × 0.1 μm, and a pinhole size of 1 μm. Z-stacks were acquired in 0.5-μm steps to image a total depth of up to 10 μm of tissue.

## Quantification of the distributions of NE-localized proteins (Fig. 2, B and G, Fig. 4 E; and Fig. S1 D)

Raw images were imported into FIJI (ImageJ; National Institutes of Health) for analysis. For Lamin Dm0 and Lamin C, their fluorescence signals were binarized to recognize nuclei and a mask image was created. The edges of nuclei were bright; therefore, to exclude this area, the mask image was eroded by five pixels. The eroded mask image area was designated a region of interest (ROI). The ROI was then superimposed onto the original image, the fluorescence intensity was measured pixel by pixel, and the standard deviation (SD) was normalized by the mean intensity of that nucleus. The macro used to determine the ROI of the nuclei was as follows:

```
run("8-bit");
run("Duplicate...," " ");
setAutoThreshold("Default dark");
//run("Threshold...");
setAutoThreshold("Huang dark");
run("Analyze Particles...," "size = 30-Infinity show = Masks exclude include");
run("Options...," "iterations = 5 count = 1 do = Erode");
run("Analyze Particles...," "size = 30-Infinity show = Outlines exclude include add");
```

Images of NPCs were slightly noisy; therefore, a median filter was used. In addition, when the NPC fluorescence signal was binarized, the difference between the bright and dark areas was so strong that the entire nucleus may not have been selected in *PIGB*-deficient larva; therefore, the binarized mask image was enlarged by five pixels to fill in the empty space and then 10 pixels were eroded to define the ROI. The macro used to determine the ROI of nuclei was as follows:

```
run("8-bit");
run("Median...," "radius = 2");
run("Duplicate...," " ");
setAutoThreshold("Mean dark");
//run("Threshold...");
run("Analyze Particles...," "size = 5-Infinity show = Masks exclude include");
run("Options...," "iterations = 5 count = 1 do = Dilate");
run("Analyze Particles...," "size = 5-Infinity show = Masks exclude include");
run("Options...," "iterations = 10 count = 1 do = Erode");
run("Analyze Particles...," "size = 5-Infinity show = Outlines exclude include add");
```

Images of Ote were also slightly noisy; therefore, a median filter was used. In addition, when the Ote fluorescence signal was binarized, the entire nucleus was not selected; therefore, Ote was costained with Lamin C and the ROI was defined using the Lamin C fluorescence signal.

## Immunoblotting

Ten carcasses of *PIGB*[27], *PIGB*[13], *PIGB*[CRP2], *PIGB*[CRP5], CS, *Lam*[K2], *LBR*[31805], Mef2-Gal4, and *PIGB*[13] larvae expressing wtPIGBmyc and PIGBmyc variants were homogenized with 100 µl of cooled lysis buffer (50 mM Tris-HCl [pH 7.5], 150 mM NaCl, 0.1% SDS,

1% Triton X-100, 1% sodium deoxycholate [192-08312; FUJIFILM Wako Chemicals], and protease inhibitor [03969-21; Nacalai Tesque]), centrifuged at 12,000 *g* for 5 min at 4°C, and the supernatants were collected. Transfected S2 and CHO cells were scraped with 200 µl of cooled lysis buffer, centrifuged at 12,000 *g* for 5 min at 4°C, and the supernatants were collected. Protein concentrations were quantified using a Pierce BCA Protein Assay Kit (23225; Thermo Fisher Scientific). Unless otherwise noted, 10 µg of protein was mixed with sample buffer (final 25 mM Tris-HCl [pH 6.8], 5% SDS, 10% glycerol, and 100 mM DTT), boiled for 5 min, and subjected to SDS-PAGE. Proteins were transferred to Immobilon-P PVDF membranes (IPVH00010; Millipore), which were blocked with PBS containing 0.05% Tween 20 (P1379; Sigma-Aldrich) and 5% skim milk (190-12865; FUJIFILM Wako Chemicals), and then incubated overnight with the appropriate primary antibodies. After three washes with PBS containing 0.05% Tween 20, the membranes were incubated with horseradish peroxidase-conjugated secondary antibodies for 3 h at room temperature. Signals were visualized with SuperSignal West PicoPlus Chemiluminescent Substrate (32132; Thermo Fisher Scientific). Images were acquired using an ImageQuant LAS4000 mini chemiluminescence detection system (GE Healthcare).

## Quantification of immunoblotting

Immunoblot quantification was performed using ImageQuant TL software (ver. 7.0; GE Healthcare).

In Fig. 2 E, lysates of 10 carcasses of *PIGB*[27] and *PIGB*[13] larvae were analyzed in three independent experiments. The intensity of each band was normalized against that of α-tubulin. Normalized values in *PIGB*[13] were compared with those in *PIGB*[27] as a control.

In Fig. 4 B, the transfection experiment was performed three times (biological replicates = 3). For each transfection, the immunoblot experiment was performed three times (experimental replicates = 3) with a standard curve, which was generated using a twofold dilution series of wtPIGBmyc-expressing cells. From the three experiments, the amounts of tubulin and PIGBmyc expressed were calculated, and the amount of PIGBmyc normalized by that of α-tubulin was averaged. The average normalized by wtPIGBmyc as a control was plotted for each transfection.

In Fig. 4 G, three batches of lysates from 10 carcasses of (Mef2), (Mef2, *PIGB*[13]), and (Mef2, *PIGB*[13] expressing wtPIGBmyc and PIGBmyc variants) larvae were prepared (biological replicates = 3). These lysates were subjected to immunoblotting three times (experimental replicates = 3) with a standard curve, which was generated using a twofold dilution series of endogenous PIGB in (Mef2). For PIGB immunoblotting, the amount of (Mef2, *PIGB*[13] expressing wtPIGBmyc [68A4]) was very high; therefore, the amount of (Mef2, *PIGB*[13] expressing wtPIGBmyc [68A4]) lysate was 1/25th of the amount of (Mef2) lysate. On the contrary, the amount of (Mef2, *PIGB*[13] expressing PIGBmyc variants) was very low; therefore, the amounts of (Mef2, *PIGB*[13] expressing 3UAS-ΔactPIGBmyc [68A4]), (Mef2, *PIGB*[13] expressing 20UAS-ΔactPIGBmyc [55C4]), and (Mef2, *PIGB*[13] expressing 3UAS-ERPIGBmyc [68A4]) lysates were eight-, two-, and

eightfold greater than the amount of (Mef2) lysate, respectively. From the three experiments, the amounts of α-tubulin and PIGBmyc expressed were calculated, and the amount of PIGBmyc normalized by the amount of α-tubulin was averaged.

### Transmission electron microscopy
Staged larvae (within 12 h of the wandering stage or 2 d later) were dissected directly in fixative (2% paraformaldehyde [3153; TAAB], 2.5% glutaraldehyde [3045; TAAB], and 150 mM sodium cacodylate, pH 7.4 [37237-35; Nacalai Tesque]) and fixed overnight at 4°C. The dissected carcasses were washed with 0.1 M phosphate buffer pH 7.4, postfixed in 1% $OsO_4$ buffered with 0.1 M phosphate buffer for 2 h, dehydrated in a graded series of ethanol, and embedded flat in Epon 812 (3402; TAAB). Ultrathin sections (70 nm thick) were collected on copper grids covered with Formvar (Nisshin EM), costained with uranyl acetate and lead citrate, and then observed by a transmission electron microscope (JEM-1400Flash; JEOL).

### FACS to measure mannosyltransferase activity of ΔactPIGBmyc (Fig. 4 A)
Transfected cells were harvested and suspended in PBS containing 0.1% BSA (0.1% BSA/PBS) and then incubated with an anti-Chinese hamster uPAR antibody for 15 min on ice. After two washes with 0.1% BSA/PBS, cells were incubated with a phycoerythrin-conjugated goat anti-mouse IgG antibody for 15 min on ice. After two washes with 0.1% BSA/PBS, cells were analyzed by a flow cytometer (FACSVerse; Becton Dickinson) using 488 and 640 nm lasers. After fluorescence compensation, $1 \times 10^4$ GFP-positive cells, which were considered to be transfected with PIGBmyc, were evaluated. A flow cytometry plot was drawn with FLOWJO software (ver. 10.8.0; FlowJo, LLC).

### DamID-seq
DamID-seq analysis was performed as described previously (Pindyurin, 2017). All DamID transgenic flies used in this study were obtained from the Bloomington *Drosophila* stock center and are listed in Table S2. The y[1] w[*]; M[w[+mC] = hs.min(FRT.STOP1)dam]ZH-51C and y[1] w[*]; M[w[+mC] = hs.min(FRT.STOP1)dam-Lam]ZH-51C strains carried DamID only and the DamID-Lamin Dm0 (Dam-Lam) fusion protein under the control of the minimal *hsp70* promotor, respectively. Insertion of a transcriptional terminator (STOP) between the promoter and Dam or Dam-Lam protein-coding sequence blocked the expression of the latter at the level of mRNA synthesis. FRT sites flanking the STOP allowed its removal in the presence of FLP recombinase. We expressed FLP recombinase using Mef2-Gal4 in a muscle-specific manner. As a result, Dam or Dam-Lam was expressed only in the muscle. The strategy used to obtain fly cross samples was as follows:

#### WT
y[1] w[*]; M[w[+mC] = hs.min(FRT.STOP1)dam]ZH-51C; Mef2-Gal4 x y[1] w[*]; P[w[+mC] = UAS-FLP.D]JD1

y[1] w[*]; M[w[+mC] = hs.min(FRT.STOP1)dam-Lam]ZH-51C; Mef2Gal4 x y[1] w[*]; P[w[+mC] = UAS-FLP.D]JD1

#### PIGB mutant (PIGB[13])
y[1] w[*]; M[w[+mC] = hs.min(FRT.STOP1)dam]ZH-51C; Mef2-Gal4, PIGB[13] x y[1] w[*]; P[w[+mC] = UAS-FLP.D]JD1; PIGB[13]

y[1] w[*]; M[w[+mC] = hs.min(FRT.STOP1)dam-Lam]ZH-51C; Mef2Gal4, PIGB[13] x y[1] w[*]; P[w[+mC] = UAS-FLP.D]JD1; PIGB[13]

Two batches of samples from 10 carcasses of WT and PIGB[13] male larvae were dissected and collected (biological replicates = 2). Genomic DNA was isolated using a DNeasy Blood and Tissue Kit (69504; Qiagen). Thereafter, 0.1 µg of genomic DNA was digested with the DpnI restriction enzyme (R0176; New England Labs) for 6 h at 37°C, ligated with DNA adaptors (2011A; TAKARA) overnight at 16°C, and digested with the DpnII restriction enzyme (R0543; New England Labs) for 1 h at 37°C. GATC-methylated fragments were amplified with an adaptor-specific primer using Advantage cDNA polymerase (Z9201N; Clontech). PCR amplification was performed on an Eppendorf Mastercycler nexus gradient thermal cycler with the following conditions: 68°C for 10 min; 94°C for 1 min; 65°C for 5 min; 68°C for 15 min; four cycles of 94°C for 1 min, 65°C for 1 min, and 68°C for 10 min; 17 cycles of 94°C for 1 min, 65°C for 1 min, and 68°C for 2 min; and hold at 15°C. High-throughput sequencing was performed on an Illumina NovaSeq 6000 sequencer, which yielded 87–130 million 150 nt paired-end reads per sample. The primers used in this study are listed in Table S3.

### DamID-seq analysis
DamID-seq analysis was performed as described previously (Leemans et al., 2019), with some modifications. The DamID adaptor sequence (5′-GGTCGCGGCCGAGGATC-3′) is present in read1 or read2 (Pindyurin, 2017). Therefore, this sequence was removed from read1 and read2 of the paired-end sequenced reads using seqkit (seqkit grep -s -r -p ^5′-GGTCGCGGCCGAGGATC-3′; Shen et al., 2016). Adapter-removed read1 and read2 were treated as single reads. This was followed by the removal of the DamID adaptor sequence (cutadapt -u 13) and the Illumina adaptor sequence (cutadapt -a 5′-GATCCTCGGCCGCGACC-3′ -m 22 -q 20; Martin, 2011). The reads were mapped to the *Drosophila* dm6 genome using Bowtie 2 (Langmead and Salzberg, 2012) with default parameters and filtered by mapping reads with quality >10. Count data of GATC sites were obtained using Samtools (Li et al., 2009) and Bedtools (Quinlan, 2014). Raw count data and 100 kb bin data were obtained. Then, $\log_2$ (Dam-Lam/Dam) was averaged across biological replicates to give the DamID score per sample, either for a 100 kb bin or binless data. LADs were defined by running a hidden Markov model over the normalized values (using the R-package HMMt; https://github.com/gui11aume/HMMt; Filion et al., 2010; Leemans et al., 2019). Pericentromere and telomere regions were obtained by referring to the cytoband definition (https://hgdownload.soe.ucsc.edu/goldenPath/dm6/database/cytoBand.txt.gz) from the UCSC genome browser (Kent et al., 2002). The calculations described above were performed using R (Kuhn et al., 2020) and R packages (readr [Wickham et al., 2018] and tidyr [Wickham et al., 2019a]), dplyr (Wickham et al., 2019b), stringr (Wickham and Wickham, 2019), ggplot2 (Wickham, 2016), UpSetR (Gehlenborg, 2019), and ggpubr (Kassambara and Kassambara, 2020).

## GO analysis

GO analysis was performed using Metascape (Zhou et al., 2019) with default parameters. For visualization, the top 10 significant GO terms were selected for each group. The observed gene ratio was also calculated.

## RT-qPCR

Three batches of samples from 20 carcasses of PIGB[27] and PIGB[13] male larvae were dissected, collected, and frozen at –80°C (biological replicates = 3). Total RNA was extracted using a PureLink RNA Mini Kit (12183018A; Thermo Fisher Scientific). SuperScript III Reverse Transcriptase (18080044; Thermo Fisher Scientific) and oligo(dT) primers were used for reverse transcription. Real-time PCR was performed on a QuantStudio 12K Flex system (Applied Biosystems) with PowerUp SYBR Green Master Mix for qPCR (A25742; Thermo Fisher Scientific). Experiments were performed with triplicates of each sample. The amount of amplified transcript was normalized against that of an internal control (rpl32). The primers used in this study are listed in Table S3.

## Measurement of nuclear mechanics

The strategy used to obtain fly cross samples was as follows:

WT; Mef2-Gal4 × UAS-mcherryNLS

PIGB mutant (PIGB[13]); Mef2-Gal4, PIGB[13]/TM6c × UAS-mcherryNLS, PIGB[13]/TM6c

Lamin Dm0 mutant (Lam[K2]); Lam[K2]/CyOG; Mef2-Gal4 × Lam[K2]/CyOG; UAS-mcherryNLS

The mechanical properties of nuclei were measured using a dual microneedle-based setup that we developed previously (Shimamoto et al., 2017). Specifically, a pair of glass microneedles (diameter: 1–2 μm), which were prepared by fabricating glass rods (G-1000; Narishige) using a capillary puller (PD-10; Narishige) and a microforge (MF-200; World Precision Instruments), was held with a three-axis hydraulic micromanipulator (MHW-3; Narishige) mounted onto the base of an inverted microscope (Ti, Nikon). The tip of one microneedle was made significantly stiff (>1,000 nN/μm) such that it could be used to apply controlled deformation to the nucleus. The tip of the other microneedle was made more flexible and stiffness-calibrated (11.3 or 12.0 nN/μm; linear range: >100 μm) such that its displacement from the equilibrium point could be used as a read-out of force that acted on it. The microscope was equipped with a 20 × dry objective lens (NA = 0.45; Nikon) and a sCMOS camera (Neo4.1; Andor). A spinning-disk confocal unit (CSU-X1; Yokogawa) and a 561-nm excitation laser (OBIS; Coherent) were used to image nuclei, and a bright-field illuminator was used to image the microneedle tips. Before the measurement, a WT, PIGB[13], or Lam[K2] larva expressing mCherry-NLS was dissected along its dorsal–ventral axis using micro-scissors and pinned to a silicon plate placed in a Petri dish such that muscle cells were facing upward when viewed sideways. The dissected larva, which had been soaked in buffer, was then placed on the microscope's motorized sample stage (MS-2000; ASI) such that a nucleus could be identified and positioned in the center of the imaging area for subsequent micromanipulation. Nuclei with a typical morphology that protruded from the upper surface of muscle cells (ventral oblique [VO] 4 or 5) were chosen because they were physically accessible with microneedles. The measurement was then started by bringing the tips of the microneedles close to the opposite sides of the nucleus until they contacted its surface via the cell membrane. After confirming there was no physical interference from the surrounding tissues or debris (by slightly moving the microneedle tips back and forth), the stiff microneedle was moved closer to the flexible microneedle to deform the nucleus while maintaining the height of the tip and then its motion was stopped after moving a given distance. Upon movement of the stiff microneedle, a resistance force developed in the nucleus, and the tip of the flexible microneedle was pushed and bent. Deflection of the tip of the flexible microneedle was monitored until the nucleus reached an equilibrium in its deformation and then the stiff microneedle was moved back to its original position to relieve the force. The measurement was repeated while changing the distance moved by the stiff microneedle each time, such that different magnitudes of deformation were applied to the same nucleus. Following this procedure, the contribution of non-nuclear structures surrounding the nucleus (e.g., cell membrane structures) to the measured force was examined. This was achieved by performing the same micromanipulation procedure at a region lacking nuclei. The measurement was performed with identical microneedle geometries (e.g., height and angle of the tip) and at a location as close as possible to the nucleus whose mechanics had been measured. During the entire measurement procedure, time-lapse images were acquired using image acquisition software (NIS-elements; Nikon) at 500-ms intervals with 200 ms exposure. The experiments were performed in a temperature-controlled room at 22 ± 1°C and completed within 1 h after dissection of each larva during which no mechanical and morphological differences were recognizable.

## Estimation of nuclear stiffness

Nuclear stiffness was determined based on the slope of the force-deformation plot, which was established from the micromanipulation measurements performed using microneedles. The amount of deformation applied was estimated in the mCherry images by measuring the change in distance between the opposite sides of the nucleus upon compression. The magnitude of the force that acted was determined in the bright-field images by measuring the displacement of the tip of the flexible microneedle from the equilibrium point and then multiplying it by the precalibrated stiffness of the tip. Image analyses were performed using ImageJ (ver. 1.48v). A force-deformation plot was generated for each nuclear sample that was subjected to cycles of compression and release with varying deformation magnitudes. Measurements were performed after nuclear deformation reached an equilibrium after each compression. The stiffness value was then obtained from the force-deformation plot by calculating the slope using linear regression. The contribution of non-nuclear materials to the measured mechanics was estimated in the same way as described above but using data acquired with no nucleus pressed between microneedles.

Statistical analysis was performed using GraphPad Prism 9 (ver. 9.1.0; GraphPad Software).

## Statistical analysis

Statistical analyses were performed as indicated in the figure legends using GraphPad Prism version 9.1.0, 9.1.1, and 10.0.0 (GraphPad Software, https://www.graphpad.com). For multiple comparisons, a one-way ANOVA was performed followed by Tukey's multiple comparison test (Fig. 2 G, Fig. 3 F; and Fig. 4, E, and G). Pairwise comparisons were performed using the paired two-tailed Student's $t$ test (Fig. 2 E) and the unpaired two-tailed Student's $t$ test (Fig. 2 B, Fig. 4, B and G, Fig. 6; and Fig. S1 D). For analyses in which parametric statistical tests were used, the normality of the data distribution was tested by the F test for the unpaired $t$ test and the Brown–Forsythe test for a one-way ANOVA with Tukey's multiple comparison test. In the nuclear mechanics experiment, P values were determined by the Mann–Whitney $U$-test (Fig. 7, E and I). In DamID experiments, P values were determined by the Wilcoxon rank sum test (Fig. 5 C). Graphs were plotted by Excel 2019 (Microsoft) and GraphPad Prism version 9.1.1 and 10.0.0 (GraphPad Software, https://www.graphpad.com).

## Online supplemental material

The DamID-seq data have been deposited in the DNA Data Bank of Japan (DDBJ) Sequence Read Archive (DRA; accession no. DRA012274). Table S1 shows the actual values, means, and SD in Fig. 6. Table S2 shows the fly stocks used in this study. Table S3 shows the primer sequences used in this study. Table S4 shows the antibodies used in this study. Fig. S1 shows the phenotype of another PIGB mutant, PIGB[CRP5], and immunoblot analysis of the mutants generated in this study. Fig. S2 shows a schematic image of the PIGB protein and the results of the statistical test in Fig. 4 E. Fig. S3 shows the correlation of DamID scores between two samples and Genome-NL interaction maps for the WT and mutant when scored using 100 kb bins. Fig. S4 shows correlation of DamID scores between two samples and Genome-NL interaction maps for the WT and mutant when scored without using bins. Fig. S5 represents an example of the Genome browser result showing DamID scores for larval wall skeletal muscle nuclei of the WT and PIGB[13] and Upset plot of the number of fLADs annotated as intron, CDS, 3UTR, 5UTR, inter_gene, and no CDS. Fig. S6 presents typical images of muscles in WT and PIGB-deficient larvae.

## Data availability

The data are available from the corresponding author upon request.

## Acknowledgments

We thank Dr. Georg Krohne (University of Würzburg, Würzburg, Germany), Dr. Pamela K. Geyer (University of Iowa, Iowa City, IA, USA), and Dr. Taroh Kinoshita (Osaka University, Osaka, Japan) for providing antibodies. We thank Dr. Pamela K. Geyer for providing fly stocks. We thank Dr. Yumiko Sakamaki of the Tokyo Medical and Dental University Research Core for her expert technical assistance with transmission electron microscopy experiments.

This work was supported by Japan Society for the Promotion of Science KAKENHI grant numbers 16H06279 (PAGS); 17H06413-1, 17H06421, and 19K06445 (S. Goto); 19K16132 (K. Kawaguchi); 22K15109 (M. Tanaka); 15K14515 and 22K18362 (Y. Shimamoto); and 20K06629, 22H00431-1, and 23K05772 (M. Yamamoto-Hino).

Author contributions: M. Yamamoto-Hino and M. Tanaka performed experiments. M. Ariura and Y.W. Iwasaki performed bioinformatics analyses of DamID-seq. M. Tanaka and Y. Shimamoto helped with the design of nuclear stiffness experiments. M. Yamamoto-Hino, K. Kawaguchi, M. Tanaka, Y. Shimamoto, and S. Goto conceived and designed the project and acquired funding. M. Yamamoto-Hino, M. Ariura, M. Tanaka, Y.W. Iwasaki, Y. Shimamoto, and S. Goto prepared the figures and wrote the paper. All authors edited or commented on the manuscript.

Disclosures: The authors declare no competing interests exist.

Submitted: 11 January 2023

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

# Supplemental material

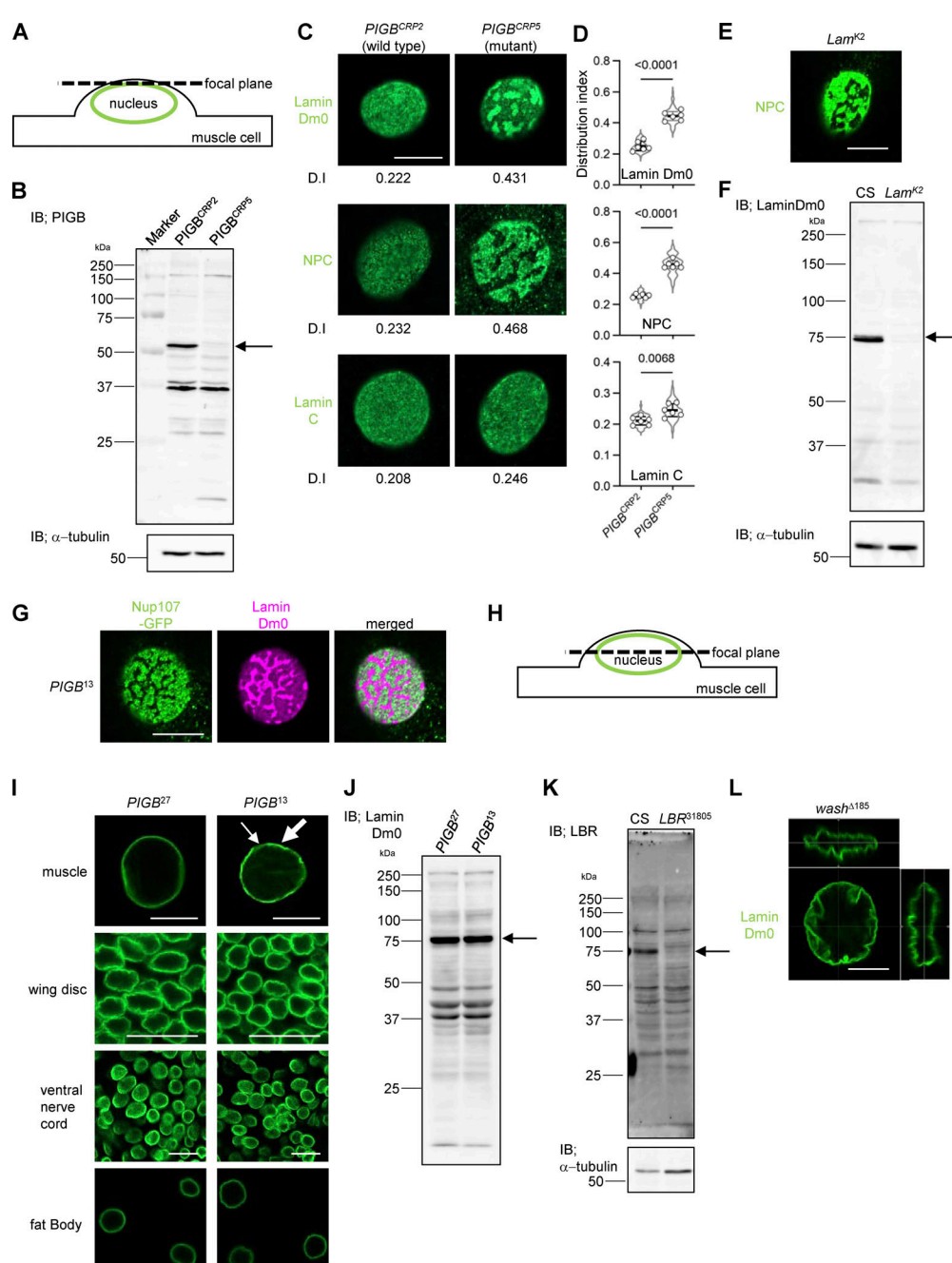

**Figure S1.** **Related to** Fig. 2**. (A)** Schematic image of a muscle nucleus with the focal plane. Green indicates the nuclear membrane, and the dotted line indicates the focal plane. **(B)** Immunoblot analysis of PIGB in larval carcasses of $PIGB^{CRP2}$ and $PIGB^{CRP5}$. The arrow indicates PIGB. α-Tubulin was used as a loading control. **(C)** Distributions of Lamin Dm0, NPCs, and Lamin C in a nucleus of larval wall skeletal muscle in $PIGB^{CRP2}$ (control) and $PIGB^{CRP5}$ (mutant). The number at the bottom of the image is the D.I. Bar, 10 μm. **(D)** Quantification of the distributions of Lamin Dm0, NPCs, and Lamin C in a nucleus of larval wall skeletal muscle in the $PIGB^{CRP2}$ and $PIGB^{CRP5}$ larvae shown in C. 10–12 nuclei per individual were measured and the average value was plotted for six individuals (white circle). The thick black horizontal bar and thin gray horizontal bar show the mean and SD of six biological replicates, respectively; >60 nuclei analyzed per strain. The superimposed violin plot shows the distribution of the D.I. The number at the top of the graph is the P value (biological replicates = 6) calculated using the unpaired two-tailed $t$ test. **(E)** Distribution of NPCs (green) in the Lamin Dm0 mutant ($Lam^{K2}$). Bar, 10 μm. **(F)** Immunoblot analysis of Lamin Dm0 in larval carcasses of CS (WT) and $Lam^{K2}$. The arrow indicates Lamin Dm0. α-Tubulin was used as a loading control. **(G)** Complementary distributions of Nup107-GFP (green) and Lamin Dm0 (magenta) in $PIGB^{13}$. Bar, 10 μm. **(H)** Schematic image of a muscle nucleus with the focal plane for observation in I. Green indicates the nuclear membrane, and the dotted line indicates the focal plane. **(I)** Distributions of Lamin Dm0 (green) in nuclei of muscle, a wing disc, a ventral nerve cord, and a fat body. In the nuclear membrane of $PIGB^{13}$ muscle cells, both dense regions (indicated by thick arrows) and sparse regions (indicated by thin arrows) of Lamin Dm0 are observed, while the distribution of Lamin Dm0 remains unchanged compared with the WT in other tissues. Bar, 10 μm. **(J)** Long-exposed and uncropped immunoblot image of Lamin Dm0 is shown in Fig. 2 D. The arrow indicates Lamin Dm0. **(K)** Immunoblot analysis of LBR in larval carcasses of the WT (CS) and LBR$^{-/-}$ ($LBR^{31805}$). The arrow indicates LBR. α-Tubulin was used as a loading control. **(L)** 3D observation of Lamin Dm0 in the $wash^{\Delta185}$ mutant. The surface of the nucleus is wrinkled, but Lamin Dm0 is distributed uniformly. Bar, 10 μm. Source data are available for this figure: SourceData FS1.

**A**

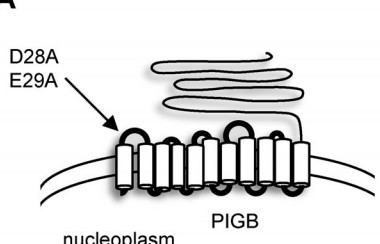

**B**

| Tukey's multiple comparisons test | | LaminDm0 | NPC |
|---|---|---|---|
| allele 1 | allele 2 | Adjusted P Value | Adjusted P Value |
| Mef2 | Mef2, PIGB[13] | <0.0001 | <0.0001 |
| Mef2 | Mef2, PIGB[13], 3UAS-wtPIGBmyc (68A4) | 0.8166 | 0.9995 |
| Mef2 | Mef2, PIGB[13], 3UAS-ΔactPIGBmyc (68A4) | 0.8756 | 0.9486 |
| Mef2 | Mef2, PIGB[13], 20UAS-ΔactPIGBmyc (55C4) | 0.9486 | 0.8994 |
| Mef2 | Mef2, PIGB[13], 3UAS-ERPIGBmyc (68A4) | <0.0001 | <0.0001 |
| Mef2, PIGB[13] | Mef2, PIGB[13], 3UAS-wtPIGBmyc (68A4) | <0.0001 | <0.0001 |
| Mef2, PIGB[13] | Mef2, PIGB[13], 3UAS-ΔactPIGBmyc (68A4) | <0.0001 | <0.0001 |
| Mef2, PIGB[13] | Mef2, PIGB[13], 20UAS-ΔactPIGBmyc (55C4) | <0.0001 | <0.0001 |
| Mef2, PIGB[13] | Mef2, PIGB[13], 3UAS-ERPIGBmyc (68A4) | 0.2727 | 0.4449 |
| Mef2, PIGB[13], 3UAS-wtPIGBmyc (68A4) | Mef2, PIGB[13], 3UAS-ΔactPIGBmyc (68A4) | >0.9999 | 0.8366 |
| Mef2, PIGB[13], 3UAS-wtPIGBmyc (68A4) | Mef2, PIGB[13], 20UAS-ΔactPIGBmyc (55C4) | 0.9992 | 0.7514 |
| Mef2, PIGB[13], 3UAS-wtPIGBmyc (68A4) | Mef2, PIGB[13], 3UAS-ERPIGBmyc (68A4) | <0.0001 | <0.0001 |
| Mef2, PIGB[13], 3UAS-ΔactPIGBmyc (68A4) | Mef2, PIGB[13], 20UAS-ΔactPIGBmyc (55C4) | >0.9999 | >0.9999 |
| Mef2, PIGB[13], 3UAS-ΔactPIGBmyc (68A4) | Mef2, PIGB[13], 3UAS-ERPIGBmyc (68A4) | <0.0001 | <0.0001 |
| Mef2, PIGB[13], 20UAS-ΔactPIGBmyc (55C4) | Mef2, PIGB[13], 3UAS-ERPIGBmyc (68A4) | <0.0001 | <0.0001 |

Figure S2.  **Related to** Fig. 4. **(A)** Membrane topology model of PIGB and the mutations in ΔactPIGB. **(B)** P values (biological replicates = 6) were calculated using a one-way ANOVA with Tukey's multiple comparison test of the distributions of Lamin Dm0 and NPCs upon expression of wtPIGBmyc and PIGBmyc variants using Mef2-Gal4 shown in Fig. 4 E.

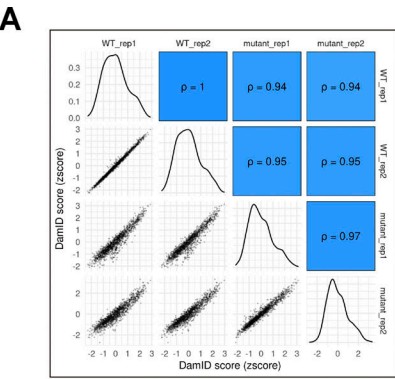

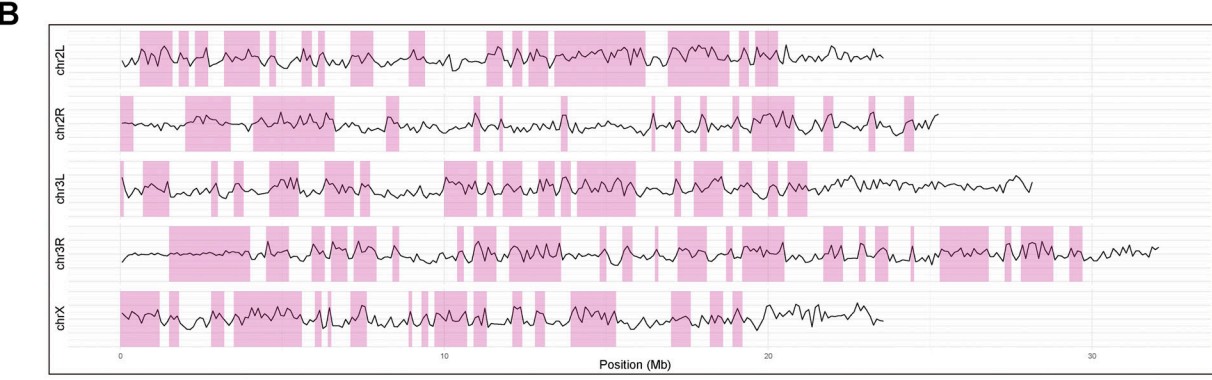

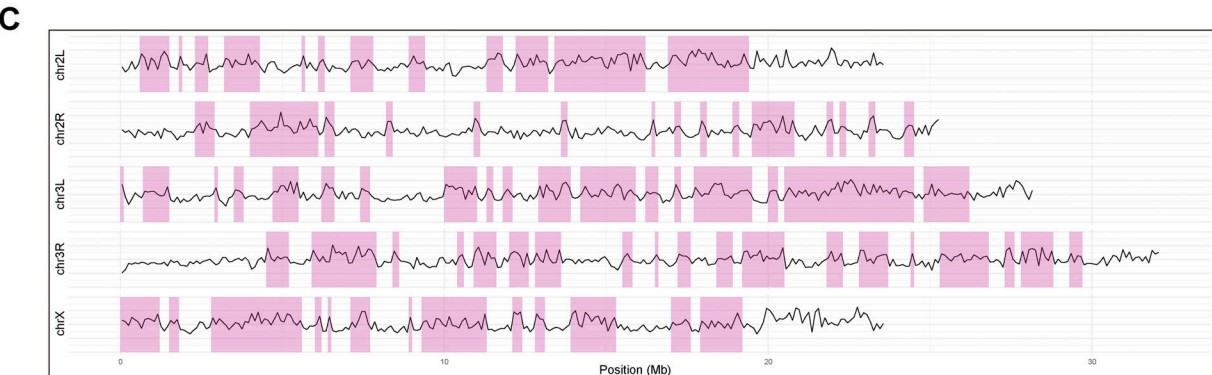

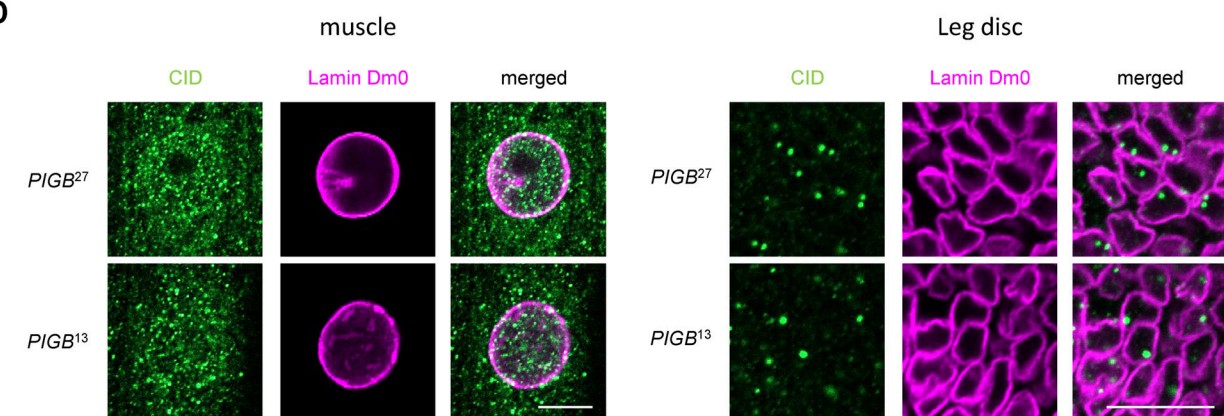

Figure S3.   **Related to** Fig. 5. **(A)** Correlation of DamID scores between two samples for the WT and mutant when scored using 100 kb bins. **(B and C)** Genome-NL interaction maps for the WT (B) and *PIGB*[13] (C) when scored using 100 kb bins. Data were obtained for 100 kb bins, and then the log$_2$ (Dam-Lam/Dam) was averaged across biological replicates to calculate the DamID score (Y-axis) for each sample. LADs shown in pink rectangles were defined by running a hidden Markov model over the normalized values (using the R-package HMMt; https://github.com/gui11aume/HMMt; Filion et al., 2010; Leemans et al., 2019). **(D)** Distributions of centromeres (green) in nuclei of muscle and leg discs. The NE was labeled with Lamin Dm0 (magenta). Bar, 10 µm.

**A**

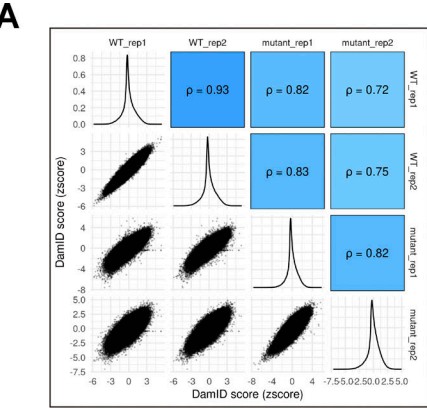

**B**

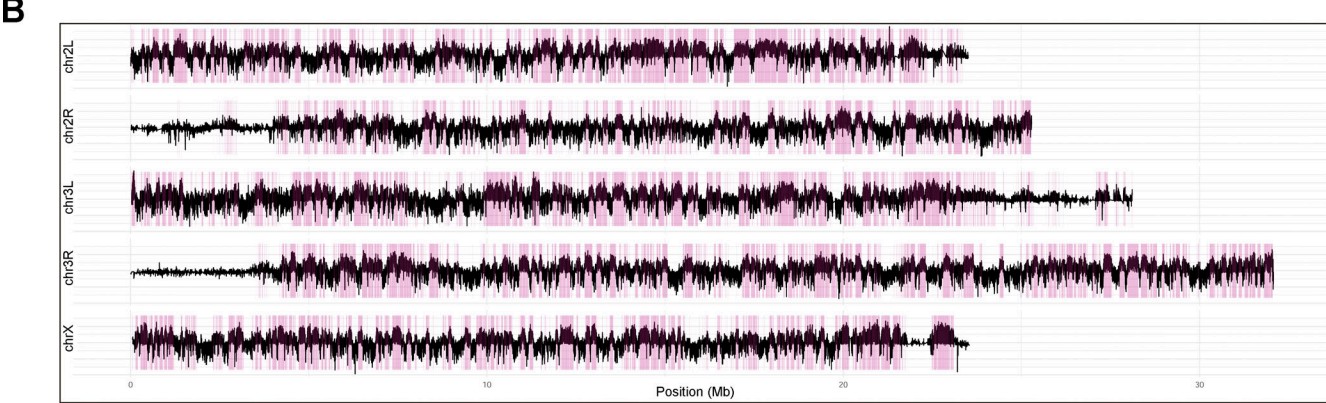

**C**

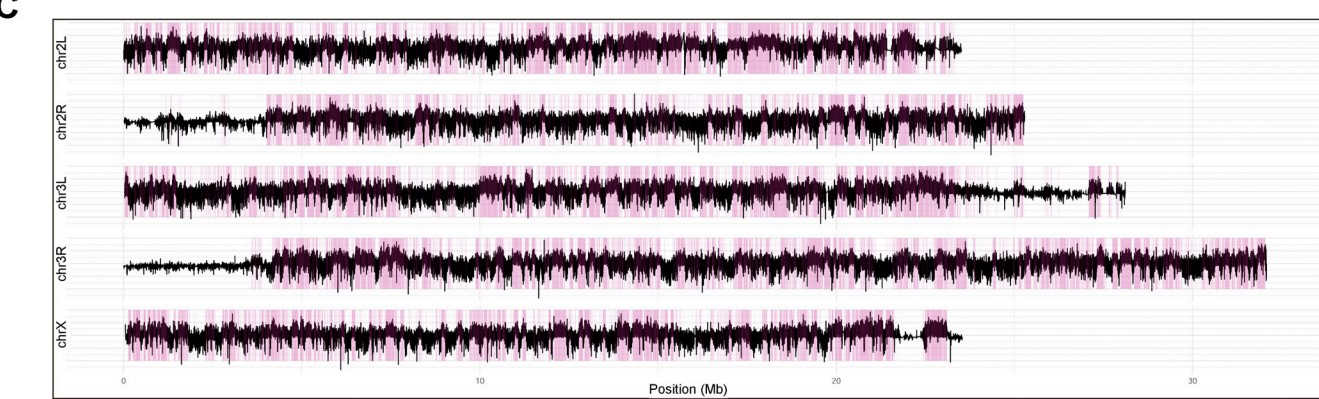

Figure S4.   **Related to** Fig. 5. **(A)** Correlation of DamID scores between two samples for the WT and mutant when scored without using bins. **(B and C)** Genome-NL interaction maps for the WT (B) and *PIGB*[13] (C) when scored without using bins. Binless data were obtained and then the $\log_2$ (Dam-Lam/Dam) was averaged across biological replicates to calculate the DamID score (Y-axis) for each sample. LADs shown in pink rectangles were defined by running a hidden Markov model over the normalized values (using the R-package HMMt; https://github.com/gui11aume/HMMt; Filion et al., 2010; Leemans et al., 2019).

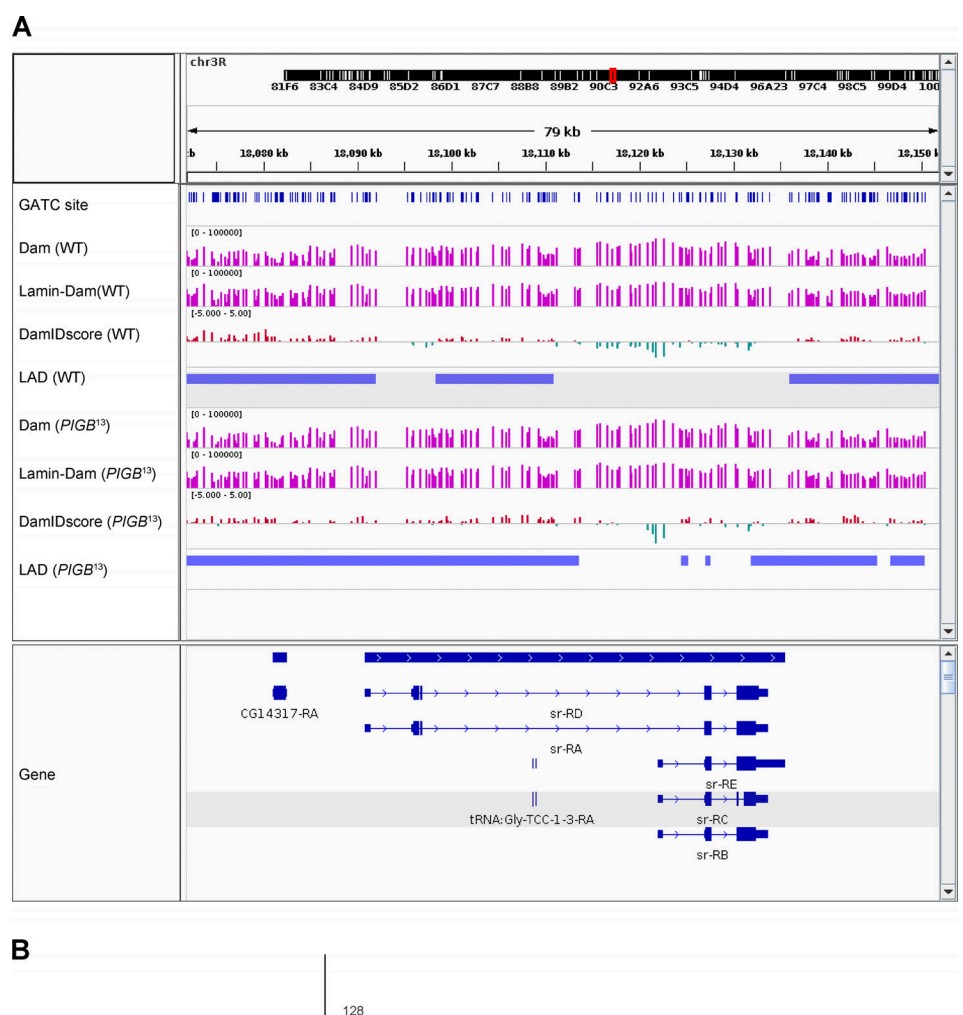

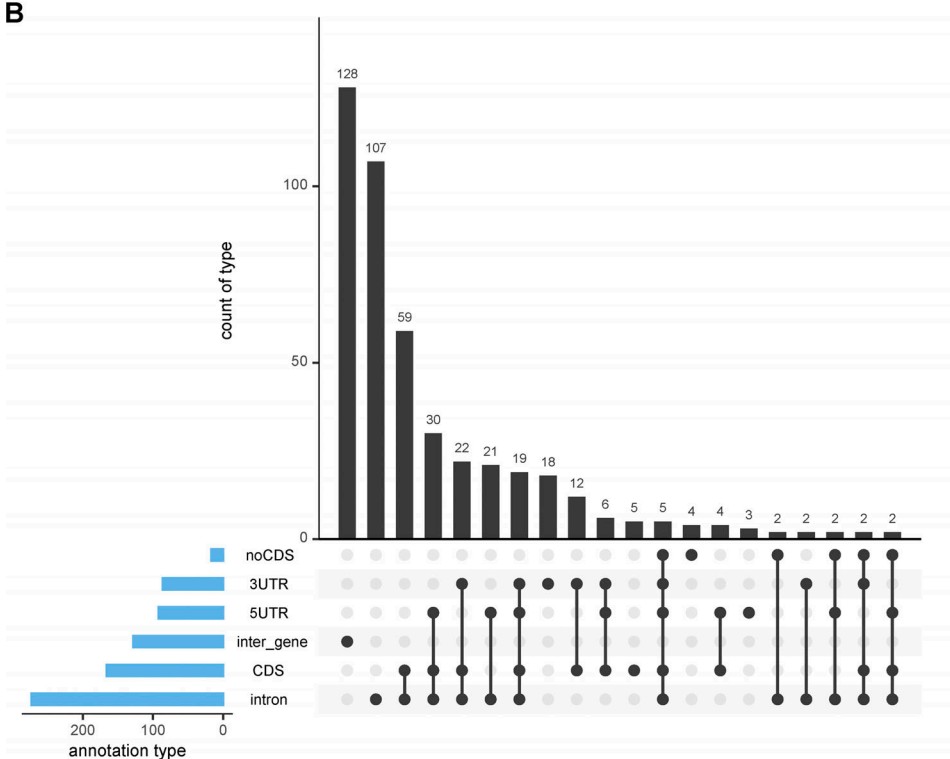

Figure S5. **Related to** Fig. 5. **(A)** Genome browser results for larval wall skeletal muscle nuclei of the WT and PIGB[13] along a 79 kb region of chromosome 3R. Y-axes depict the log$_2$-transformed Dam-Lam to Dam-only methylation ratio. Rectangles below each map represent calculated LADs. **(B)** Upset plot of the number of fLADs annotated as intron, CDS, 3UTR, 5UTR, inter_gene, and no CDS.

Figure S6.   **Related to** Fig. 8. Typical images of phalloidin-stained larval wall muscle in *PIGB*[27] [<12 h] (top left three images), *PIGB*[13] [>2 d] (top right three images), *PIGB*[CRP2] [<12 h] (bottom left three images), and *PIGB*[CRP5] [>2 d] (bottom right three images). Arrows indicate cracks in VL1 in *PIGB*-deficient larva. Bar, 500 μm.

**Provided online are Table S1, Table S2, Table S3, and Table S4. Table S1 shows the actual values, means, and SD in** Fig. 6. **Table S2 shows the fly stocks used in this study. Table S3 shows the primer sequences used in this study. Table S4 shows the antibodies used in this study.**

