## [Peer Review File · The Journal of Cell Biology]

PIGB maintains nuclear lamina organization in skeletal muscle of *Drosophila*

Miki Yamamoto-Hino, Masaru Ariura, Masahito Tanaka, Yuka Iwasaki, Kohei Kawaguchi, Yuta Shimamoto, and Satoshi Goto

Corresponding Author(s): Satoshi Goto, Rikkyo University

Review Timeline:

Submission Date:	2023-01-11
Editorial Decision:	2023-03-27
Revision Received:	2023-10-09
Editorial Decision:	2023-10-27
Revision Received:	2023-11-15

Monitoring Editor: Bas van Steensel

Scientific Editor: Dan Simon

Transaction Report:

DOI: <https://doi.org/10.1083/jcb.202301062>

March 27, 2023

Re: JCB manuscript #202301062

Prof. Satoshi Goto
Rikkyo University
Department of Life Science
3-34-1 Nishi-Ikebukuro
Toshima-ku
Tokyo 1718501
Japan

Dear Prof. Goto,

Thank you for submitting your manuscript entitled "PIGB maintains nuclear lamina organization in skeletal muscle of *Drosophila*." Your manuscript has been assessed by two expert reviewers, whose comments are appended below. We apologize for the delay and thank you for your patience with the peer review process. Although the reviewers express potential interest in this work, significant concerns unfortunately preclude publication of the current version of the manuscript in JCB.

You will see that Reviewer #1 asks for additional evidence that PIGB enzymatic activity is not required for its regulation of lamin Dm0 localization. Reviewer #2 notes that only a single PIG-B mutant background was used and since this may have other mutations the findings need to be confirmed by repeating experiments in a heteroallelic PIG-B mutant background as the reviewer suggests or by generating a new null mutant with CRISPR editing. Other important requests are missing data and method descriptions, controls for LAD assays, confirmation of lamin protein absence in Lam-K2 mutant muscle. Both reviewers also ask for data quantifications and text changes to clarify results and add discussion. We agree that all of these are important points that have to be addressed in full.

Regarding statistical analysis please note the following JCB policies:

Error bars on graphic representations of numerical data must be clearly described in the figure legend. The number of independent data points (n) represented in a graph must be indicated in the legend. Please, indicate whether 'n' refers to technical or biological replicates (i.e. number of analyzed cells, samples or animals, number of independent experiments). If independent experiments with multiple biological replicates have been performed, we strongly recommend using distribution-reproducibility SuperPlots (please see Lord et al., JCB 2020) to better display the distribution of the entire dataset, and report statistics (such as means, error bars, and P values) that address the reproducibility of the findings.

Statistical methods should be explained in full in the materials and methods. For figures presenting pooled data the statistical measure should be defined in the figure legends. Please also be sure to indicate the statistical tests used in each of your experiments (both in the figure legend itself and in a separate methods section) as well as the parameters of the test (for example, if you ran a t-test, please indicate if it was one- or two-sided, etc.). Also, if you used parametric tests, please indicate if the data distribution was tested for normality (and if so, how). If not, you must state something to the effect that "Data distribution was assumed to be normal but this was not formally tested."

Please let us know if you are able to address the major issues outlined above and wish to submit a revised manuscript to JCB. Note that a substantial amount of additional experimental data likely would be needed to satisfactorily address the concerns of the reviewers. The typical timeframe for revisions is three to four months. While most universities and institutes have reopened labs and allowed researchers to begin working at nearly pre-pandemic levels, we at JCB realize that the lingering effects of the COVID-19 pandemic may still be impacting some aspects of your work, including the acquisition of equipment and reagents. Therefore, if you anticipate any difficulties in meeting this aforementioned revision time limit, please contact us and we can work with you to find an appropriate time frame for resubmission. Please note that papers are generally considered through only one revision cycle, so any revised manuscript will likely be either accepted or rejected.

If you choose to revise and resubmit your manuscript, please also attend to the following editorial points. Please direct any editorial questions to the journal office.

GENERAL GUIDELINES:

Text limits: Character count is < 40,000, not including spaces. Count includes title page, abstract, introduction, results, discussion, and acknowledgments. Count does not include materials and methods, figure legends, references, tables, or supplemental legends.

Figures: Your manuscript may have up to 10 main text figures. To avoid delays in production, figures must be prepared according to the policies outlined in our Instructions to Authors, under Data Presentation,

<https://jcb.rupress.org/site/misc/ifora.xhtml>. All figures in accepted manuscripts will be screened prior to publication.

IMPORTANT: It is JCB policy that if requested, original data images must be made available. Failure to provide original images upon request will result in unavoidable delays in publication. Please ensure that you have access to all original microscopy and blot data images before submitting your revision.

Supplemental information: There are strict limits on the allowable amount of supplemental data. Your manuscript may have up to 5 supplemental figures. Up to 10 supplemental videos or flash animations are allowed. A summary of all supplemental material should appear at the end of the Materials and methods section.

Please note that JCB now requires authors to submit Source Data used to generate figures containing gels and Western blots with all revised manuscripts. This Source Data consists of fully uncropped and unprocessed images for each gel/blot displayed in the main and supplemental figures. Since your paper includes cropped gel and/or blot images, please be sure to provide one Source Data file for each figure that contains gels and/or blots along with your revised manuscript files. File names for Source Data figures should be alphanumeric without any spaces or special characters (i.e., SourceDataF#, where F# refers to the associated main figure number or SourceDataFS# for those associated with Supplementary figures). The lanes of the gels/blots should be labeled as they are in the associated figure, the place where cropping was applied should be marked (with a box), and molecular weight/size standards should be labeled wherever possible. Source Data files will be made available to reviewers during evaluation of revised manuscripts and, if your paper is eventually published in JCB, the files will be directly linked to specific figures in the published article.

If you choose to resubmit, please include a cover letter addressing the reviewers' comments point by point. Please also highlight all changes in the text of the manuscript.

Regardless of how you choose to proceed, we hope that the comments below will prove constructive as your work progresses. We would be happy to discuss them further once you've had a chance to consider the points raised. You can contact the journal office with any questions, cellbio@rockefeller.edu or call (212) 327-8588.

Thank you for thinking of JCB as an appropriate place to publish your work.

Sincerely,

Bas van Steensel, PhD
Monitoring Editor
Journal of Cell Biology

Dan Simon, PhD
Scientific Editor
Journal of Cell Biology

Reviewer #1 (Comments to the Authors (Required)):

The authors, Yamamoto-Hino et al. previously reported that lamin B is required for localization of PIGB in the inner nuclear membrane, and is essential for biosynthesis of glycosylphosphatidylinositol (GPI)-anchored proteins. Conversely, the manuscript reports that PIGB is prerequisite for nuclear lamina (NL) organization through correct distribution of nuclear lamin B (lamin Dm0) in *Drosophila*. The authors found that loss of PIGB causes disorganization of NL. PIGB is an enzyme involved in GPI biosynthesis, but the enzymatic function is not required for the NL organization, suggesting that PIGB is a multifunctional protein. Deletion of PIGB affected lamina-associated domains (LADs), forming smaller LADs. In addition, loss of PIGB resulted in less rigidity of the nuclei. The authors carefully designed the experiments and analyzed the data. The paper is well written and the results look solid. However, several critical points need to be addressed.

1) The authors claimed that PIGB affects the localization of lamin B, which is independent of PIGB enzymatic activity. However, it was concluded from the result based on only one PIGB mutant (Δ -act PIGBmyc). They previously compared human and *Drosophila* PIGB sequences and produced a PIGB chimera localized to the endoplasmic reticulum. Can this chimera rescue the NL phenotypes observed in PIGB mutant? In addition, when mammalian PIGB is expressed, cell surface expression of uPAR is restored, but NL phenotypes are not rescued?

2) The results in Figure 2A, 2H and Figure 3D are not quantitative at all. The reviewer understands that the authors show a representative pictures in WT and PIGB mutant larvae. However, it is not possible to judge the results from the single figure. The

reviewer requests for the authors to provide the readers quantitative data.

3) In Figure 2M and N, actin filaments are observed in nucleus in PIGB mutant larvae. Explain the reason why the actin localization is affected when the NL organization is impaired.

4) In Figure 3A, expression of PIGBmyc appears to completely rescue cell surface expression of uPAR. Is it the result of stable or transient expression cells? Clarify it in the figure legend.

5) In Figure 5, the PIGB mutation appears to have less nuclear stiffness than the Lamin mutation. Can the authors explain or discuss the reason?

Reviewer #2 (Comments to the Authors (Required)):

Summary

In this manuscript by Yamamoto-Hino et al., the authors report the role of PIG-B on the structure and function of the lamin network in *Drosophila*. PIG-B is an enzyme involved in glycosylphosphatidylinositol (GPI) synthesis. The authors examine effects of loss of PIG-B on 1) distribution of the *Drosophila* B-type lamin (lamin Dm0) and other nuclear lamina proteins; 2) test whether mutations in other genes encoding other nuclear lamina components alter lamin Dm0 distribution; 3) evaluate whether PIG-B enzymatic activity is required for lamin Dm0 distribution and 4) examine effects of PIG-B loss on muscle structure, nuclear organization of LADs and nuclear stiffness.

Comments

This is an interesting study that presented data to support a role for PIG-B in the establishment or maintenance of the *Drosophila* B-type lamin at the nuclear periphery. Clear effects of increased and decreased levels of PIG-B are shown, and the evidence is strong that mutations in genes encoding other nuclear lamina components do not change Lamin Dm0 localization. It is interesting that PIG-B13 mutants disrupt lamin Dm0 differently than Lamin C, reinforcing our understanding that these lamins form separate networks. Also, it is surprising that the disrupted network of lamin Dm0 differs from the disrupted network of NUPs, indicating great complexity of inner nuclear membrane networks. Much of the further investigation is focused on third instar muscle structure and function in PIG-B13 mutants. These are important studies, but these data are less convincing. Critically, the authors only conduct experiments on one PIG-B13 mutant background, making it unclear if the observed effects result from PIG-B loss or second site mutations on the PIG-B13 chromosome. This is a serious concern, as PIG-B13 is a lethal allele, requiring that this allele is carried over a balancer chromosome. In such cases, the second site mutations are frequently accumulated, which can confound interpretation. This is because all of their studies not only test homozygous effects of PIG-B13, but also test homozygous effects of all other mutant alleles on the chromosome. It is critical that the authors repeat many of their experiments in a heteroallelic PIG-B mutant background. They could take advantage of existing resources in the community, such as small deletion chromosomes. Second, the *mef2>PIG-B* rescue studies are concerning, as *mef2> actPIG-B* rescues Lamin Dm0 distribution without producing detectable levels of protein (Fig. 3E). It is unclear how a non-enzymatic function of this enzyme can rescue the distribution with vanishingly small amounts of protein. In fact, the authors fail to quantify the levels of the produced proteins. It is possible that differences in genetic background between the two expressor backgrounds is responsible for PIG-B loss or the observed effects do not result from the PIG-B13 mutation. Of note, the authors fail to indicate whether the UAS- *actPIG-B* and UAS- *actPIG-B* transgenes are located in the genome. Did they use the PhiC31 integration system or are transgenes randomly distributed? If random insertions are used, this intensifies the concern of background effects. Extension of these studies to include a second PIG-B mutant background is essential.

Major Concerns:

1. Several questions arise from the phalloidin staining of PIG-B13 mutant muscle. First, although phalloidin stripes are found near the nucleus, it is unclear if they are truly inside the nucleus. This point needs clarification. Second, why do these data suggest that PIG-B13 mutants affect Lamin C function. It should be noted that Ote localization is aberrant in PIG-B13 mutants and Ote is an ortholog of emerin, a nuclear lamina component shown to regulate actin (see PMID: 15773747). Third, the transmission electron micrographs could resolve this issue. Were actin filaments found in the PIG-B13 mutants? Fourth, more information is needed to evaluate the EM pictures, as no structural landmarks are provided to allow insight that the same area of the cell and the same cell type is shown. Fifth, age effects were not assessed on the wild type animals.
2. Western data in Fig. 3 needs to be quantified. The authors state that the same levels of PIG-B myc and *actPIG-B myc* are produced in CHO cells, but the blot is not convincing. Further, all western analysis lack quantification, which is needed.
3. In Figs. 3F and 6B,C, quantification of normal versus abnormal is presented. However, the authors fail to adequately describe how abnormal was scored. Pictures representing the two classes would be helpful.
4. In studies of the effects of PIG-B on formation of LADs, there is no description of these studies in the materials and methods. How was Dam-Lam expressed in a muscle-specific manner using the FLP-FRT system? Further, the motivation between comparing binned data and non-binned data is unclear. Thresholds for calling LADs are unclear. Critically, data presented in Fig. S3, show pink boxes (LAD calls) around regions that are flat (WT, 3R). In addition, Fig. S3 data do not show LADs at centromeres, whereas data in Fig. 4B suggest that there are changes in LADs at centromeres.
5. Studies validating the Dam-Lam studies are preliminary and lack controls. It is unclear why the authors state that they focused

on 128 genes belonging to muscle function when they only tested 10. Further, data in Fig. S3I require controls. The authors need to quantify expression from an equal number of genes outside of predicted LADs to understand if these differences in the muscle genes are a consequence of LAD differences or indirect.

6. In Fig. 5G, tests of the deformability of nuclei in muscles dissected from LamK2 mutants are shown. However, the authors do not show how and when Lamin Dm0 levels are affected in the LamK2 mutants. Of note, lamins are stable proteins and the LamK2 mutants have maternally provided protein that can perdure into larval stages. The authors need to complete western analysis of mutant muscle to ensure that they are analyzing cells that lack Lamin Dm0.

7. In the discussion, the authors indicate that they tested if loss of PIG-B affected post translational modifications of Lamin Dm0. Although they state that immunoblotting found no difference, these data are not shown. These data should be included in the manuscript.

Minor comments:

1. The authors need to clarify how they conclude that over-expression of PIG-B leads to aggregates of Lamin Dm0 that are continuous with the INM (in first results section).

2. Materials and methods are incomplete. For example, a complete description of the molecular lesions associated with each mutation is needed.

3. The author include several references of "data not shown". These either need to be removed or the data must be included.

4. References are incomplete, as several investigations of effects of lamin mutant backgrounds on larval muscle function were not included.

5. In the discussion, one header reads "PIG-B contributes to nuclear membrane integrity", but integrity was not assessed.

Dr. Bas van Steensel,
Monitoring Editor
Journal of Cell Biology

Ref: #202301062

October 9th, 2023

Dear Dr. van Steensel,

We greatly appreciate the very helpful suggestions of the reviewers and editor, which have enabled us to improve the manuscript. We have fully addressed all of the reviewers' concerns and performed several new experiments. I hope you will find the revised manuscript suitable for publication in *Journal of Cell Biology*.

Our detailed responses to each of the referees' comments are provided below.

Yours sincerely,

Satoshi Goto, PhD

Responses to Reviewer #1:

We appreciate the critical and helpful suggestions. We have addressed all the concerns and conducted additional experiments using endoplasmic reticulum-localized PIGB (ERPIGB) to demonstrate that the nuclear localization of PIGB, rather than its enzymatic activity, is necessary to regulate the localization of Lamin Dm0. We have provided detailed responses to each of the comments below.

1) The authors claimed that PIGB affects the localization of lamin B, which is independent of PIGB enzymatic activity. However, it was concluded from the result based on only one PIGB mutant (delta-act PIGBmyc). They previously compared human and Drosophila PIGB sequences and produced a PIGB chimera localized to the endoplasmic reticulum. Can this chimera rescue the NL phenotypes observed in PIGB mutant?

RE: We addressed this comment by performing additional experiments using ERPIGB. ERPIGB was unable to rescue the heterogeneous localizations of Lamin Dm0 and nuclear pore complexes (NPCs) (Fig. 3D and 3E). This suggests strongly that localization of PIGB to the nuclear envelope (NE), rather than its enzymatic activity, is essential for formation and maintenance of a well-organized nuclear lamina (NL).

In addition, when mammalian PIGB is expressed, cell surface expression of uPAR is restored, but NL phenotypes are not rescued?

RE: The experiment could not be conducted because mammalian PIGB is scarcely expressed in *Drosophila* (Yamamoto-Hino *et al.*, J Cell Sci., 2018).

2) The results in Figure 2A, 2H and Figure 3D are not quantitative at all. The reviewer understands that the authors show a representative pictures in WT and PIGB mutant larvae. However, it is not possible to judge the results from the single figure. The reviewer requests for the authors to provide the readers quantitative data.

RE: We quantified the heterogeneity in the distribution of Lamin Dm0 by assessing variations in the signal intensity of Lamin Dm0. We introduced the Distribution Index (D.I.), which represents the intensity variation normalized by the mean fluorescence intensity of nuclear images. The numerical values displayed at the bottom of the images in Fig. 2A, 2F, S1C, and 3D correspond to the D.I. We captured images of 10–12 nuclei per individual, calculated the D.I., and compared the mean D.I. among six individuals. The results of these statistical analyses are presented in Fig. 2B, S1D, and 3E. The quantitative methods and statistical procedures are described in detail in the respective figure legends and the Materials and Methods section.

3) In Figure 2M and N, actin filaments are observed in nucleus in PIGB mutant larvae. Explain the reason why the actin localization is affected when the NL organization is impaired.

RE: Our findings illustrated in Fig. 2A, 2B, S1C, and S1D demonstrate abnormalities in the distributions of Lamin C and the LEM-domain protein Ote resulting from PIGB depletion. Previous studies indicated that human Lamins A and B bind directly to purified actin in vitro (Simon *et. al.*, Nucleus, 2014), and *lamin C* null mutant and *lamin C* Δ N-expressing

Drosophila larvae display actin polymers in nuclei (Dialynas et al., Development, 2010, Schulze et al., PLoS One, 2009). Additionally, in mammals, it has been reported that Emerin, a LEM-domain protein, directly binds to nuclear actin, regulating processes such as mRNA transcription and chromatin remodeling (Wilson et al., Nuclear Organization in Development and Disease, 2005). These findings imply that loss of PIGB would impact potential functions of Lamin C and LEM-domain proteins, including Ote, in regulation of nuclear actin polymerization and formation or stabilization of a cortical actin network in the nucleoplasm. This has been additionally described in the revised manuscript.

4) In Figure 3A, expression of PIGBmyc appears to completely rescue cell surface expression of uPAR. Is it the result of stable or transient expression cells? Clarify it in the figure legend.

RE: We transiently co-transfected cells with a PIGBmyc expression plasmid and an EGFP expression plasmid at a molecular ratio of 4:1. The GFP-positive cell population, which is indicative of successful transfection with the PIGBmyc construct, was assessed. This is described in the respective figure legend and the Materials and Methods section.

5) In Figure 5, the PIGB mutation appears to have less nuclear stiffness than the Lamin mutation. Can the authors explain or discuss the reason?

RE: Comparing the nuclear stiffness of PIGB and lamin mutants was not possible because they were not measured side-by-side on the same day. Our current method for measuring nuclear stiffness is time-consuming and allows us to collect data for only one sample type and its corresponding control at a time. Consequently, conducting simultaneous measurements for the PIGB mutant, the *Lamin Dm0* mutant, and their respective control samples present a challenge.

Responses to Reviewer #2:

We appreciate the critical and helpful suggestions. We have addressed all the concerns and conducted additional experiments using a new PIGB mutant to eliminate the possibility that the observed phenotypes are caused by the genetic background. We have provided detailed responses below.

Comments

This is an interesting study that presented data to support a role for PIG-B in the establishment or maintenance of the *Drosophila* B-type lamin at the nuclear periphery. Clear effects of increased and decreased levels of PIG-B are shown, and the evidence is strong that mutations in genes encoding other nuclear lamina components do not change Lamin Dm0 localization. It is interesting that PIG-B13 mutants disrupt lamin Dm0 differently than Lamin C, reinforcing our understanding that these lamins form separate networks. Also, it is surprising that the disrupted network of lamin Dm0 differs from the disrupted network of NUPs, indicating great complexity of inner nuclear membrane networks. Much of the further investigation is focused on third instar muscle structure and function in PIG-B13 mutants. These are important studies, but these data are less convincing. Critically, the authors only conduct experiments on one PIG-B13 mutant background, making it unclear if the observed effects result from PIG-B loss or second site mutations on the PIG-B13 chromosome. This is a serious concern, as PIG-B13 is a lethal allele, requiring that this allele is carried over a balancer chromosome. In such cases, the second site mutations are frequently accumulated, which can confound interpretation. This is because all of their studies not only test homozygous effects of PIG-B13, but also test homozygous effects of all other mutant alleles on the chromosome. It is critical that the authors repeat many of their experiments in a heteroallelic PIG-B mutant background. They could take advantage of existing resources in the community, such as small deletion chromosomes.

RE: To confirm that the phenotype is attributable to the PIGB deletion, we created another null mutant using CRISPR editing. This mutant, known as *PIGB*^{CRP5}, carries a 5-bp deletion spanning from nucleotide 74 to nucleotide 78 relative to the first ATG of the *PIGB* gene, resulting in a protein of only 27 amino acids. Similar to *PIGB*¹³, this mutant larva exhibited mortality during the late larval stage. *PIGB*^{CRP2}, which was generated concurrently, does not possess the mutation and served as the wild-type (WT) control. We verified the absence of PIGB protein expression in larval muscle of *PIGB*^{CRP5} using western blotting (Fig. S1B). We conducted a series of experiments to investigate the distributions of NE-localized proteins (Fig. S1C and D), the presence of actin fibers in the nucleus (Fig. S1M), alterations in gene expression (Fig. S3J), and deformations in muscle structure (Fig. 6B and Fig. S4) in *PIGB*^{CRP5}. Nearly all phenotypes observed in *PIGB*¹³ were also observed in *PIGB*^{CRP5}. This suggests that the observed phenotypes were a consequence of PIGB depletion rather than being attributable to a second-site mutation. This is described in the corresponding regions and the Materials and Methods section of the revised manuscript.

Second, the *mef2>PIG-B* rescue studies are concerning, as *mef2> Δ actPIG-B* rescues Lamin Dm0 distribution without producing detectable levels of protein (Fig. 3E). It is unclear how a non-enzymatic function of this enzyme can rescue the distribution with vanishingly small amounts of protein. In fact, the authors fail to quantify the levels of the produced proteins. It is possible that differences in genetic background between the two expressor backgrounds is responsible for PIG-B loss or the observed effects do not result from the PIG-B13 mutation. Of note, the authors fail to indicate whether the UAS- Δ actPIG-B and UAS- Δ actPIG-B transgenes are located in the genome. Did they use the PhiC31 integration system or are transgenes randomly distributed? If random insertions are used, this intensifies the concern of background effects. Extension of these studies to include a second PIG-B mutant background is essential.

RE: Immunoblot analysis revealed that expression of 3UAS- Δ actPIGBmyc (68A4) in our previous study was only 2.6% of the endogenous PIGB level (Fig. 3F and G). Remarkably, even such modest expression of PIGB was sufficient for formation and maintenance of the highly organized NL. In this study, wtPIGBmyc and PIGBmyc variants were expressed using the Gal4-UAS system. This system involves crossing the Mef2-Gal4 driver, which expresses the Gal4 transcription factor in a muscle-specific pattern, to a responder possessing a transgene driven by a UAS element. The UAS element is bound by Gal4, which results in activation of transgene expression. A higher number of UAS elements is expected to increase expression.

It is essential to note that in our previous study, both 3UAS-wtPIGBmyc and 3UAS- Δ actPIGBmyc were integrated into the same genomic locus, namely, 68A4, using the PhiC31 integration system.

In our revised study, we aimed to enhance the expression level and eliminate positional effects of the inserted transgene by generating a new Δ actPIGBmyc allele, 20UAS- Δ actPIGBmyc, which was integrated into the 55C4 genomic locus. This alteration resulted in 20UAS- Δ actPIGBmyc (55C4) expression at 25% of the endogenous PIGB level (Fig. 3F and G). Importantly, this modification rescued the heterogeneous distributions of Lamin Dm0 and NPCs (Fig. 3D and E), accumulation of nuclear actin fibers (Fig. 3H), and muscle deformation (Fig. 6C).

Furthermore, we explored the potential rescue effects of ERPIGB (3UAS-ERPIGBmyc) based on the recommendation of Reviewer #1. Notably, 3UAS-ERPIGBmyc was integrated into the same genomic locus (68A4) as 3UAS-wtPIGBmyc and 3UAS- Δ actPIGBmyc. However, expression of ERPIGBmyc failed to rescue the heterogeneous distributions of Lamin Dm0 and NPCs, suggesting that insertion of the transgene into 68A4 did not rescue

the observed phenotypes. Additionally, it is noteworthy that the expression level of 3UAS-ERPIGBmyc was 15% of the endogenous PIGB level, which was higher than that of 3UAS- Δ actPIGBmyc (68A4) (Fig. 3F and G). This observation suggests that the inability of 3UAS-ERPIGBmyc to rescue the phenotypes was not due to low expression levels.

We included information regarding creation of the new lineage in the Materials and Methods section. The results of experiments utilizing the new lineage are presented in Fig. 3D–H and 6C. The insertion sites for each lineage are indicated in the respective figures.

Major Concerns:

1. Several questions arise from the phalloidin staining of PIG-B13 mutant muscle. First, although phalloidin stripes are found near the nucleus, it is unclear if they are truly inside the nucleus. This point needs clarification.

RE: To confirm the presence of actin fibers in the nucleus, we conducted 3D observations by labeling the NE with Lamin C. As a result, we observed signals of phalloidin in the nucleus, indicating that actin localized inside, rather than outside, the nucleus (Fig. S1N).

Second, why do these data suggest that PIG-B13 mutants affects Lamin C function. It should be noted that Ote localization is aberrant in PIG-B13 mutants and Ote is an ortholog of emerin, a nuclear lamina component shown to regulate actin (see PMID: 15773747).

RE: In Fig. 2A, 2B, S1C, and S1D, PIGB depletion led to abnormal distributions of Lamin C and the LEM-domain protein Ote. Human Lamins A and B bind directly to purified actin *in vitro* (Simon et al., Nucleus, 2014), and *Lamin C* null mutant and *Lamin C* Δ N-expressing *Drosophila* larvae display actin polymers in nuclei (Dialynas et al., Development, 2010, Schulze et al., PLoS One, 2009). Furthermore, in mammals, it has been reported that Emerin, a LEM-domain protein, directly binds to nuclear actin, regulating processes such as mRNA transcription and chromatin remodeling (Wilson et al., Nuclear Organization in Development and Disease, 2005). These findings imply that loss of PIGB may impact potential functions of Lamin C and LEM-domain proteins, including Ote, in regulation of nuclear actin polymerization and formation or stabilization of a cortical actin network in the NL. This is described in the revised manuscript.

Third, the transmission electron micrographs could resolve this issue. Were actin filaments found in the PIG-B13 mutants?

RE: Muscle nuclei lack thickness and are spaced far apart, making it difficult to observe numerous nuclei. In this study, we were unable to identify nuclei containing actin filaments using electron microscopy.

Fourth, more information is needed to evaluate the EM pictures, as no structural landmarks are provided to allow insight that the same area of the cell and the the same cell type is shown.

RE: As described in the Materials and Methods section, we prepared larval carcasses and generated sections from the surface corresponding to the inner side of the larva. Muscle nuclei are positioned at the top of muscle cells; therefore, nuclei appearing first in the sections are presumed to belong to muscle cells. We observed these nuclei. In Fig. S1O, we have indicated rough ER and mitochondria with arrows as landmarks.

Fifth, ages effects were not assessed on the wild type animals.

RE: WT wandering third instar larvae undergo pupation within 12 h, making it impossible to sample individuals that have spent an extended period at the larval stage. For the observation of nuclear actin and muscle damage, we have included descriptions in the respective part of the text and figures for individuals sampled within 12 h as well as those for which more than 2 days had elapsed for both WT and mutant types (Fig. 2H, 2I, S1M–P, 3H, 6, and S4).

2. Western data in Fig. 3 needs to be quantified. The authors state that the same levels of PIG-B myc and Δ actPIG-B myc are produced in CHO cells, but the blot is not convincing. Further, all western analysis lack quantification, which is needed.

RE: The transfection experiment was conducted three times (biological replicates = 3). In each transfection, the immunoblot experiment was performed three times (experimental replicates = 3). From these three experiments, the quantities of tubulin and PIGBmyc expression were determined, and the quantity of PIGBmyc normalized by the quantity of tubulin was calculated and averaged. The average, normalized to wtPIGBmyc as a control, was plotted for each transfection. Statistical analysis was performed using the unpaired two-tailed t-test with three biological replicates.

Quantitative results are presented in the main text and Fig. 3B. Additional explanations have been added to the respective figure legends and the Materials and Methods section.

3. In Figs. 3F and 6B,C, quantification of normal versus abnormal is presented. However, the authors fail to adequately describe how abnormal was scored. Pictures representing the two classes would be helpful.

RE: For the assessment of nuclear actin, individuals were categorized as 'normal' when no actin fibers were observed in the nucleus and 'abnormal' when nuclear actin fibers were visible. Twelve individuals were observed for comparison of the WT and mutant (Fig. S1M), and 36 individuals were examined in the rescue experiment (Fig. 3H). Further explanations have been added to the respective figure legends.

Regarding muscle damage, individuals were categorized as 'normal' when no muscle damage was detected and 'abnormal' when muscle damage was observed. Twelve individuals were assessed for comparison of the WT and mutant (Fig. 6B), and 36 individuals were examined in the rescue experiment (Fig. 6C). Additional explanations have been included in the respective figure legends. For a representative example of muscle damage, please refer to Fig. S4.

4. In studies of the effects of PIG-B on formation of LADs, there is no description of these studies in the materials and methods. How was Dam-Lam expressed in a muscle-specific manner using the FLP-FRT system?

RE: DamID-seq analysis was performed as described previously (Pindyurin et al., 2017). All DamID transgenic flies used in this study were obtained from the Bloomington *Drosophila* stock center and are listed in Table S1. The $y[1] \ w[*]; M\{w[+mC]=hs.min(FRT.STOP1)dam\}ZH-51C$ and $y[1] \ w[*]; M\{w[+mC]=hs.min(FRT.STOP1)dam-Lam\}ZH-51C$ strains carried DamID only and the DamID-Lamin Dm0 (Dam-Lam) fusion protein under the control of the minimal *hsp70* promoter, respectively. Insertion of a transcriptional terminator (STOP) between the promoter and Dam or Dam-Lam protein-coding sequence blocked expression of the latter at the level of mRNA synthesis. FRT sites flanking the STOP allowed its removal in the presence of FLP recombinase. We expressed FLP recombinase using Mef2-Gal4 in a muscle-specific manner. As a result, Dam or Dam-Lam was expressed only in muscle. The strategy used to obtain fly cross samples was as follows:

WT

$y[1] \ w[*]; M\{w[+mC]=hs.min(FRT.STOP1)dam\}ZH-51C; Mef2Gal4 \times y[1] \ w[*];$

P{w[+mC]=UAS-FLP.D}JD1
y[1] w[*]; M{w[+mC]=hs.min(FRT.STOP1)dam-Lam}ZH-51C; Mef2Gal4 x y[1] w[*];
P{w[+mC]=UAS-FLP.D}JD1

PIGB mutant (*PIGB*¹³)

y[1] w[*]; M{w[+mC]=hs.min(FRT.STOP1)dam}ZH-51C; Mef2Gal4, *PIGB*¹³ x y[1] w[*];
P{w[+mC]=UAS-FLP.D}JD1; *PIGB*¹³

y[1] w[*]; M{w[+mC]=hs.min(FRT.STOP1)dam-Lam}ZH-51C; Mef2Gal4, *PIGB*¹³ x y[1] w[*];
P{w[+mC]=UAS-FLP.D}JD1; *PIGB*¹³

These details have been included in the Materials and Methods section.

Further, the motivation between comparing binned data and non-binned data is unclear.

RE: Binning can cause the LAD score variations to become less discernible. However, choosing not to use binning allows a more detailed comparison of LAD score fluctuations. This motivated our decision to explore this approach.

Thresholds for calling LADs are unclear.

RE: We identified LADs by utilizing the R-package Hidden Markov Model (HMMt, available at <https://github.com/gui11aume/HMMt>), which was developed by Filion et al. (Cell, 2010, <https://pubmed.ncbi.nlm.nih.gov/20888037/>) and Leemans et al. (Cell, 2019, <https://pubmed.ncbi.nlm.nih.gov/30982597/>). Importantly, this approach did not necessitate the establishment of specific thresholds. Additional explanations have been included in the respective text, the figure legends, and the Materials and Methods section.

Critically, data presented in Fig. S3, show pink boxes (LAD calls) around regions that are flat (WT, 3R). In addition, Fig. S3 data do not show LADs at centromeres, whereas data in Fig. 4B suggest that there are changes in LADs at centromeres.

RE: We apologize for any confusion regarding the content of Fig. 4B. The Y-axis represents DamID scores, not LADs. We have updated the legend of Fig. 4B to clarify this.

5. Studies validating the Dam-Lam studies are preliminary and lack controls. It is unclear why the authors state that they focused on 128 genes belonging to muscle function when they only tested 10.

RE: We conducted an analysis focusing on the top 10 highly expressed genes out of the 128 genes. Genes with low expression levels were excluded from the analysis due to their tendency to exhibit greater data variability. In Fig. S3J, we confirmed that similar gene expression changes occurred in the *PIGB*^{CRP5} and *PIGB*¹³ mutants.

Further, data in Fig. S3I require controls. The authors need to quantify expression from an equal number of genes outside of predicted LADs to understand if these differences in the muscle genes are a consequence of LAD differences or indirect.

RE: The changes in gene expression due to PIGB depletion involve two factors: one is alteration of LADs and the other is loss of GPI-anchored proteins. Therefore, it is possible that gene expression changes may occur even in genes that are not part of LADs, making them unsuitable as controls. In our analysis, we acknowledge the possibility of a mixture of changes driven by alterations of LADs and loss of GPI-anchored proteins among the genes showing altered expression. We have discussed this possibility in the Discussion section.

6. In Fig. 5G, tests of the deformability of nuclei in muscles dissected from LamK2 mutants are shown. However, the authors do not show how and when Lamin Dm0 levels are affected in the LamK2 mutants. Of note, lamins are stable proteins and the LamK2 mutants have maternally provided protein that can perdure into larval stages. The authors need to complete western analysis of mutant muscle to ensure that they are analyzing cells that lack Lamin Dm0.

RE: We conducted western blot analysis of *Lam*^{K2} carcasses and confirmed that Lamin Dm0 protein expression was below the detection limit (Fig. S1F).

7. In the discussion, the authors indicate that they tested if loss of PIG-B affected post translational modifications of Lamin Dm0. Although they state that immunoblotting found no difference, these data are not shown. These data should be included in the manuscript.

RE: We have added uncropped, long-exposure images of the Lamin Dm0 immunoblot for *PIGB*²⁷ and *PIGB*¹³ in Fig. S1J.

Minor comments:

1. The authors need to clarify how they conclude that over-expression of PIG-B leads to

aggregates of Lamin Dm0 that are continuous with the INM (in first results section).

RE: When observing *Mef2>wtPIGBmyc* in the 3D image in Fig. 1B, the lamin signal in the nucleoplasm, which was visible in the X-Y section, appeared not to be separated from the NE signal when viewed in the X-Z section. Furthermore, the nuclear lamin aggregated on the NE and appeared to invaginate.

2. Materials and methods are incomplete. For example, a complete description of the molecular lesions associated with each mutation is needed.

RE: We have added detailed information for each mutant used in Table S1.

3. The author include several references of "data not shown". These either need to be removed or the data must be included.

RE: We have included Fig. S3D to illustrate the position of centromeres, which was previously indicated as 'data not shown'. Additionally, we have provided Fig. S1I to depict the distributions of Lamin Dm0 in fat bodies, nerves, and the wing disc, which was also previously mentioned as 'data not shown'.

4. References are incomplete, as several investigations of effects of lamin mutant backgrounds on larval muscle function were not included.

RE: We have added a discussion regarding the observed phenotypes of *Lamin Dm0* mutants, including reduced locomotion, the absence or slight reduction in size of the LO1 muscle, and fine structural defects in fibrillar organization (Muñoz-Alarcón, PLoS One, 2007). Additionally, we have included information in the Discussion section about expression of Lamin C with an N-terminal deletion and mutated Lamin C found in laminopathies in muscle cells, which has been reported to result in lethality, mislocalization of nuclear membrane proteins, nucleo-cytoskeletal uncoupling, and locomotion defects (Schulze, PLoS One, 2009, Dialynas, Development, 2010, Dialynas, Hum Mol Genet., 2012, Zwerger, Hum Mol Genet., 2013, Coombs, Redox Biol., 2021, and Shaw, Front Cell Dev Biol., 2022). Relevant references have been cited.

5. In the discussion, one header reads "PIG-B contributes to nuclear membrane integrity", but integrity was not assessed.

RE: We have revised 'integrity' to 'stiffness' in one of the headers in the Discussion section.

October 27, 2023

RE: JCB Manuscript #202301062R

Prof. Satoshi Goto
Rikkyo University
Department of Life Science
3-34-1 Nishi-Ikebukuro
Toshima-ku
Tokyo 1718501
Japan

Dear Prof. Goto:

Thank you for submitting your revised manuscript entitled "PIGB maintains nuclear lamina organization in skeletal muscle of *Drosophila*." The manuscript was assessed by original Reviewer #1. Reviewer #2 was not available to re-review. We would be happy to publish your paper in JCB pending final revisions necessary to meet our formatting guidelines (see details below). Please also consider adding to your discussion the recently published PMID: 37433644.

A. MANUSCRIPT ORGANIZATION AND FORMATTING:

1) Text limits: Character count for Articles is < 40,000, not including spaces. Count includes title page, abstract, introduction, results, discussion, and acknowledgments. Count does not include materials and methods, figure legends, references, tables, or supplemental legends.

2) Figure formatting: Articles may have up to 10 main text figures. Scale bars must be present on all microscopy images, including inset magnifications. Molecular weight or nucleic acid size markers must be included on all gel electrophoresis. Please avoid pairing red and green for images and graphs to ensure legibility for color-blind readers. If red and green are paired for images, please ensure that the particular red and green hues used in micrographs are distinctive with any of the colorblind types. If not, please modify colors accordingly or provide separate images of the individual channels.

3) Statistical analysis: Error bars on graphic representations of numerical data must be clearly described in the figure legend. The number of independent data points (n) represented in a graph must be indicated in the legend. Please, indicate whether 'n' refers to technical or biological replicates (i.e. number of analyzed cells, samples or animals, number of independent experiments). If independent experiments with multiple biological replicates have been performed, we recommend using distribution-reproducibility SuperPlots (please see Lord et al., JCB 2020) to better display the distribution of the entire dataset, and report statistics (such as means, error bars, and P values) that address the reproducibility of the findings.

Statistical methods should be explained in full in the materials and methods. For figures presenting pooled data the statistical measure should be defined in the figure legends. Please also be sure to indicate the statistical tests used in each of your experiments (both in the figure legend itself and in a separate methods section) as well as the parameters of the test (for example, if you ran a t-test, please indicate if it was one- or two-sided, etc.). Also, if you used parametric tests, please indicate if the data distribution was tested for normality (and if so, how). If not, you must state something to the effect that "Data distribution was assumed to be normal but this was not formally tested."

4) Materials and methods: Should be comprehensive and not simply reference a previous publication for details on how an experiment was performed. Please provide full descriptions (at least in brief) in the text for readers who may not have access to referenced manuscripts. The text should not refer to methods "...as previously described."

5) For all cell lines, vectors, constructs/cDNAs, etc. - all genetic material: please include database / vendor ID (e.g., Addgene, ATCC, etc.) or if unavailable, please briefly describe their basic genetic features, even if described in other published work or gifted to you by other investigators (and provide references where appropriate). Please be sure to provide the sequences for all of your oligos: primers, si/shRNA, RNAi, gRNAs, etc. in the materials and methods. You must also indicate in the methods the source, species, and catalog numbers/vendor identifiers (where appropriate) for all of your antibodies, including secondary. If antibodies are not commercial, please add a reference citation if possible.

6) Microscope image acquisition: The following information must be provided about the acquisition and processing of images:

- Make and model of microscope
- Type, magnification, and numerical aperture of the objective lenses
- Temperature
- Imaging medium
- Fluorochromes
- Camera make and model
- Acquisition software
- Any software used for image processing subsequent to data acquisition. Please include details and types of operations involved (e.g., type of deconvolution, 3D reconstitutions, surface or volume rendering, gamma adjustments, etc.).

7) References: There is no limit to the number of references cited in a manuscript. References should be cited parenthetically in the text by author and year of publication. Abbreviate the names of journals according to PubMed.

8) Supplemental materials: Articles generally have up to 5 supplemental figures and 10 videos. Figures cannot span multiple pages. Currently your supplemental figures 1 & 3 are both longer than one page. Please either condense and consolidate these data into single page figures or split them up into separate figures. We will be able to give you the extra space if needed.

9) eTOC summary: A ~40-50 word summary that describes the context and significance of the findings for a general readership should be included on the title page. The statement should be written in the present tense and refer to the work in the third person. It should begin with "First author name(s) et al..." to match our preferred style.

10) Conflict of interest statement: JCB requires inclusion of a statement in the acknowledgements regarding competing financial interests. If no competing financial interests exist, please include the following statement: "The authors declare no competing financial interests." If competing interests are declared, please follow your statement of these competing interests with the following statement: "The authors declare no further competing financial interests."

11) A separate author contribution section is required following the Acknowledgments in all research manuscripts. All authors should be mentioned and designated by their first and middle initials and full surnames. We encourage use of the CRediT nomenclature (<https://casrai.org/credit/>).

12) ORCID IDs: ORCID IDs are unique identifiers allowing researchers to create a record of their various scholarly contributions in a single place. Please note that ORCID IDs are required for all authors. At resubmission of your final files, please be sure to provide your ORCID ID and those of all co-authors.

13) JCB requires authors to submit Source Data used to generate figures containing gels and Western blots with all revised manuscripts. This Source Data consists of fully uncropped and unprocessed images for each gel/blot displayed in the main and supplemental figures. Since your paper includes cropped gel and/or blot images, please be sure to provide one Source Data file for each figure that contains gels and/or blots along with your revised manuscript files. File names for Source Data figures should be alphanumeric without any spaces or special characters (i.e., SourceDataF#, where F# refers to the associated main figure number or SourceDataFS# for those associated with Supplementary figures). The lanes of the gels/blots should be labeled as they are in the associated figure, the place where cropping was applied should be marked (with a box), and molecular weight/size standards should be labeled wherever possible. Source Data files will be directly linked to specific figures in the published article.

14) Journal of Cell Biology now requires a data availability statement for all research article submissions. These statements will be published in the article directly above the Acknowledgments. The statement should address all data underlying the research presented in the manuscript. Please visit the JCB instructions for authors for guidelines and examples of statements at (<https://rupress.org/jcb/pages/editorial-policies#data-availability-statement>).

B. FINAL FILES:

-- Cover images: If you have any striking images related to this story, we would be happy to consider them for inclusion on the

journal cover. Submitted images may also be chosen for highlighting on the journal table of contents or JCB homepage carousel. Images should be uploaded as TIFF or EPS files and must be at least 300 dpi resolution.

****It is JCB policy that if requested, original data images must be made available to the editors. Failure to provide original images upon request will result in unavoidable delays in publication. Please ensure that you have access to all original data images prior to final submission.****

****The license to publish form must be signed before your manuscript can be sent to production. A link to the electronic license to publish form will be sent to the corresponding author only. Please take a moment to check your funder requirements before choosing the appropriate license.****

Thank you for this interesting contribution, we look forward to publishing your paper in Journal of Cell Biology.

Sincerely,

Bas van Steensel, PhD
Monitoring Editor
Journal of Cell Biology

Dan Simon, PhD
Scientific Editor
Journal of Cell Biology

Reviewer #1 (Comments to the Authors (Required)):

The authors properly addressed the reviewers' concerns. The reviewer recommends to publish the manuscript in JCB.